# Multi-Scale Finetuning for Encoder-based Time Series Foundation Models

**Zhongzheng Qiao**[1,2,3]  **Chenghao Liu**[4]  **Yiming Zhang**[1]  **Ming Jin**[5]  **Quang Pham**[4]

**Qingsong Wen**[6]  **P.N.Suganthan**[7]  **Xudong Jiang**[1]  **Savitha Ramasamy**[2,3]

## Abstract

Time series foundation models (TSFMs) demonstrate impressive zero-shot performance for time series forecasting. However, an important yet underexplored challenge is how to effectively finetune TSFMs on specific downstream tasks. While naive finetuning can yield performance gains, we argue that it falls short of fully leveraging TSFMs' capabilities, often resulting in overfitting and suboptimal performance. Given the diverse temporal patterns across sampling scales and the inherent multi-scale forecasting capabilities of TSFMs, we adopt a causal perspective to analyze finetuning process, through which we highlight the critical importance of explicitly modeling multiple scales and reveal the shortcomings of naive approaches. Focusing on *encoder-based* TSFMs, we propose **M**ulti**S**cale **F**ine**T**uning (**MSFT**), a simple yet general framework that explicitly integrates multi-scale modeling into the finetuning process. Experimental results on three different backbones (MOIRAI, MOMENT and UNITS) demonstrate that TSFMs finetuned with MSFT not only outperform naive and typical parameter efficient finetuning methods but also surpass state-of-the-art deep learning methods. Codes are available at https://github.com/zqiao11/MSFT.

## 1 Introduction

Time series foundation models (TSFMs) have emerged as a transformative direction within the time series forecasting (TSF) community [2, 43, 8]. By pretraining on extensive time series datasets, these models possess universal knowledge, enabling them to achieve impressive zero-shot performance on various forecasting tasks. Despite significant advancements in TSFM research, current studies predominantly focus on model pretraining and zero-shot evaluation, while paying limited attention to the critical challenge of effectively finetuning these universal models for specific downstream tasks. In contrast, finetuning pretrained models has become the standard pipeline for real-world applications in domains such as natural language processing (NLP) and computer vision (CV). Research in these fields has revealed key challenges in finetuning foundation models, including preserving pretrained knowledge [24], avoiding overfitting [15], and ensuring efficient adaptation [13, 48].

Existing finetuning strategies for TSFMs often rely on naive approaches, such as full finetuning or linear probing [2, 12, 11]. While these methods may offer performance gains, we argue that **naive finetuning is suboptimal for TSFMs** as it fails to account for the intrinsic *multi-scale properties* of both time series data and TSFMs. As a data modality generated from continuous real-world processes, time series are inherently entangled and can be decomposed across multiple scales [23, 18]. A time series can exhibit distinct temporal patterns at different sampling scales. For instance, as shown in

---

[1]Nanyang Technological University. [2]Institute for Infocomm Research, A*STAR. [3]CNRS@CREATE. [4]Salesforce AI Research. [5]Griffith University. [6]Squirrel Ai Learning. [7]Qatar University. Mail to: Zhongzheng Qiao <qiao0020@e.ntu.edu.sg>, Chenghao Liu <chenghao.liu@salesforce.com>.

39th Conference on Neural Information Processing Systems (NeurIPS 2025).

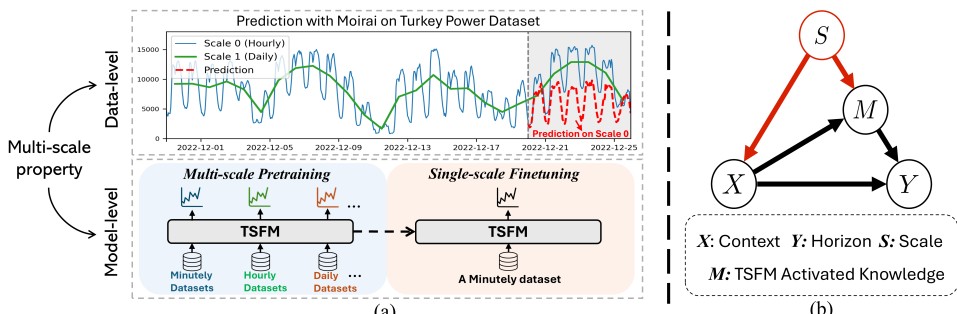

(a)                                              (b)

Figure 1: (a) Multi-scale property in time series foundation model (TSFM) finetuning. Finetuning TSFMs on the original scale may overlook potential temporal patterns in time series and underutilize their multi-scale forecasting capabilities learned during pretraining. (b) Causal graph for forecasting of TSFMs. Nodes denote the abstract data variables and directed edges denote the causality, i.e. cause → effect. Scale $S$ acts as a confounder, influencing both input context series $X$ and model's activated knowledge $M$ (shown in red).

Figure 1 (a), energy consumption measured at the hour level shows micro-scopic local usage patterns, whereas daily records suppress these finer details, highlighting macro-scopic consumption trends instead. This multi-scale nature poses additional challenges, as naive finetuning tends to overfit the model to patterns at the original scale, overlooking the latent dynamics that prevail at coarser scales. From a modeling perspective, TSFMs pretrained on extensive, multi-scale datasets are inherently equipped with robust multi-scale forecasting capabilities. However, naive finetuning fails to harness this potential, as it restricts learning to the original scale. Consequently, it underutilizes the pretrained knowledge of TSFMs, capturing only partial temporal patterns. Such failure not only limits the generalizability of TSFMs across scales but also leads to suboptimal downstream performance.

To address the aforementioned challenge, we begin by analyzing the finetuning process of TSFMs through a causal lens. The relationship among key variables is shown in Figure 1(b). Specifically, the objective of finetuning is to adapt the model $P(Y|X)$ to capture temporal patterns and better predict the horizon $Y$ given the context $X$. However, the presence of scale $S$ as a confounder introduces spurious correlations between context $X$ and the knowledge $M$ activated within TSFM, causing the model to rely on correlations that lack causal grounding. Directly forecast with $P(Y|X)$ would mistakenly associate non-causal but positively correlated context $X$ to horizon $Y$. To overcome this, we propose using the interventional distribution $P(Y|do(X))$, which isolates the true causal effect of $X$ on $Y$ by blocking the influence of the confounder $S$. We will elaborate on how this is achieved through backdoor adjustment [27] in Section 3.

This causal perspective highlights the need for explicitly modeling multiple scales during TSFM finetuning. However, integrating multi-scale modeling in this context remains underexplored and presents several non-trivial challenges—despite its success in standard time series forecasting modeling [34, 42, 41]. **First**, most TSFMs tokenize time series through patching [25], resulting in tokens at different scales exhibiting varying resolutions and temporal dynamics. This discrepancy complicates the finetuning of the unified input projection and attention weights. **Second**, applying attention across multi-scale tokens can introduce spurious dependencies due to misaligned time indices, making it difficult to capture true temporal relationships. Thus, the attention mechanism must account for or bypass index-related biases. **Finally**, since the model produces separate predictions at each scale, effectively aggregating these multi-scale outputs is essential for accurate and robust forecasting.

To close the gap, we propose a novel encoder-based TSFM finetuning framework using multi-scale modeling, namely **MSFT**. Our contributions are summarized as follows:

1. Building on causal insights, we identify the limitations of naive finetuning for TSFMs and propose a multi-scale modeling approach for TSFM finetuning. To the best of our knowledge, this is the first work to introduce multi-scale modeling into TSFMs.

2. We propose MSFT, a simple yet effective finetuning framework for encoder-based TSFMs. MSFT begins by downsampling time series into multiple scales and independently tokenizing each scale at its own resolution. Scale-specific modules are applied to the input projection and attention layers to activate scale-specific knowledge. Decoupled dependency modeling is then performed on the concatenated multi-scale sequence, enabling the model to capture both within-scale (via

in-scale attention) and cross-scale (via cross-scale aggregator) dependencies. Finally, a learnable weighting strategy is employed to aggregate the multi-scale prediction results.

3. Our extensive evaluation on various datasets for Long Sequence Forecasting [44] and Probabilistic Forecasting [43] demonstrates that **MSFT** not only significantly improves the fintuning results of TSFMs but also surpasses other state-of-the-art models trained from scratch.

## 2   Preliminaries

**Problem Formulation.**   We first define the TSF task, in which the model predicts a horizon window given a context window. Let $C$ denote the context length and $H$ the horizon length. Context window $\mathbf{X} \in \mathbb{R}^{C \times D}$ and horizon window $\mathbf{Y} \in \mathbb{R}^{H \times D}$ are consecutively extracted from the same time series $\mathbf{x}_{1:T} = (\mathbf{x}_1, \mathbf{x}_2, \ldots, \mathbf{x}_T)$, where $D$ is the feature dimension at each time step. The sample at time step $t$ is denoted as $(\mathbf{X}_t, \mathbf{Y}_t)$, where $\mathbf{X}_t = (\mathbf{x}_{t-C}, \ldots, \mathbf{x}_{t-1})$ and $\mathbf{Y}_t = (\mathbf{x}_t, \ldots, \mathbf{x}_{t+H-1})$. Given a model parameterized by $\theta$ and a training dataset $\mathcal{D}^{\text{train}} = \{(\mathbf{X}_t, \mathbf{Y}_t)\}_{t=1}^{T_o}$, the objective is to learn the model parameter $\theta^*$ to achieve minimum error on the testing set $\mathcal{D}^{\text{test}} = \{(\mathbf{X}_t, \mathbf{Y}_t)\}_{t=T_o+1}^{T}$.

**Multi-Scale Generation.**   In *multi-scale modeling* (see Appendix A.2 for the detailed definition of this concept), the standard approach for generating multi-scale sequences is based on *average pooling* [34, 42]. Given a training sample $(\mathbf{X}, \mathbf{Y})$, both context and horizon windows are downsampled into multiple temporal scales using non-overlapping average pooling. Specifically, downsampling factor is commonly set to 2, resulting in a set of scales defined by $1, 2, \ldots, 2^K$, where $K$ is the number of downsampled scales. Let $\mathcal{S}$ denote the set of multi-scale time series as $\mathcal{S} = \{\mathbf{S}_0, \ldots, \mathbf{S}_K\}$, where $\mathbf{S}_i = (\mathbf{X}^i, \mathbf{Y}^i)$ corresponds to the $i$-th scale series, formed by concatenating the downsampled context $\mathbf{X}^i \in \mathbb{R}^{C_i \times D}$ and downsampled horizon $\mathbf{Y}^i \in \mathbb{R}^{H_i \times D}$. Here, $C_i = \lceil \frac{C}{2^i} \rceil$ and $H_i = \lceil \frac{H}{2^i} \rceil$. Note that $\mathbf{S}_0$ represents the input series at the original scale.

**Encoder-based TSFM.**   We outline the architectural framework of existing encoder-based TSFMs [43, 12, 11] from a high-level perspective. These models adopt an encoder-only Transformer [39] architecture and segment univariate time series into a sequence of patch tokens [25]. While multivariate extensions are supported in some models [43, 11], we focus on the univariate case for illustration ($D = 1$), without loss of generality. The pretraining is conducted by masked reconstruction [9]. Given a time series $(\mathbf{X}, \mathbf{Y})$, the series is segmented into non-overlapping patch tokens of size $P$, resulting in a sequence of patches $\boldsymbol{x} \in \mathbb{R}^{N \times P}$, where $N = \lceil \frac{C}{P} \rceil + \lceil \frac{H}{P} \rceil$. The goal is to forecast the predictive horizon by $\hat{\mathbf{Y}} = f_\theta(\boldsymbol{x})$, where $f_\theta$ is a transformer with the block number $L$ and model dimension $d$. Specifically, Equation 1 represents the procedure of calculating $\hat{\mathbf{Y}} = f_\theta(\boldsymbol{x})$:

$$\boldsymbol{h}^0 = \text{InProject}(\boldsymbol{x}); \quad \boldsymbol{h}^l = \text{AttnBlock}(\boldsymbol{h}^{l-1}), \, l = 1, ..., L; \quad \hat{\mathbf{Y}} = \text{OutProject}(\boldsymbol{h}^L) \qquad (1)$$

Let $\boldsymbol{h}^l \in \mathbb{R}^{N \times d}$ represent the token embeddings produced by layer $l$. The input projection $\text{InProject}$ embeds patch tokens into input embeddings $\boldsymbol{h}^0$. Each $\text{AttnBlock}$ consists of a multi-head self-attention layer, followed by a feed-forward network (FFN) and normalization layers. The output projection $\text{OutProject}$ maps the output embeddings $\boldsymbol{h}^L$ to the prediction $\hat{\mathbf{Y}}$, either directly [12, 11] or indirectly by first producing distributional parameters from which $\hat{\mathbf{Y}}$ is sampled [43]. We summarize the architectural features and training losses of each model in Appendix B.3.

## 3   Multi-Scale Finetuning of TSFM

### 3.1   Multi-Scale Effect on TSFM: A Causal View

As we discussed in Section 1, both time series data and TSFMs exhibit multi-scale properties. We take scale into account during TSFM finetuning and construct a Structural Causal Model (SCM) [28] as illustrated Figure 1 (b). The nodes denote the abstract data variables, and the directed edges denote the causality, i.e., cause $\rightarrow$ effect. Denoting input context window data as $X$, scale as $S$, and prediction horizon window data as $Y$, we discuss the rationale for each link as follows:

$\boldsymbol{X \leftarrow S}$. Given an observed recording of the context period, the input context series $X$ is directly influenced by the scale $S$. Although corresponding to the same temporal range, $X$ exhibits different temporal patterns and resolutions at different sampling rates.

$\boldsymbol{S \rightarrow M \leftarrow X}$. We denote $M$ as the activated knowledge within the pretrained TSFM's knowledge space, conditioned on input context. $S \rightarrow M$ indicates that the scale of data activates the correspond-

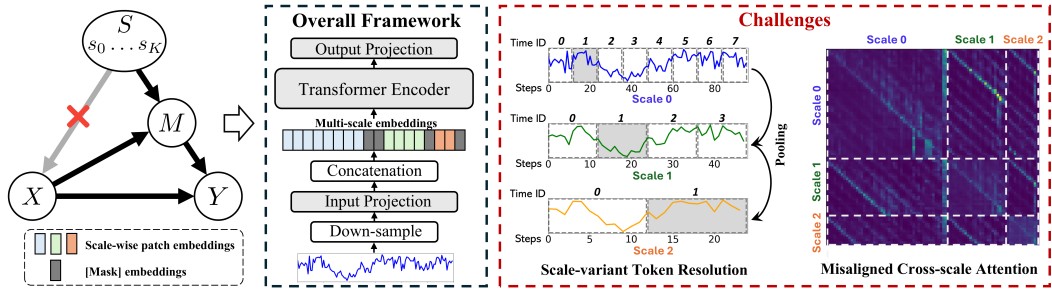

Figure 2: (a): The intervened Structural Causal Models (SCM) and overall **MultiScale FineTuning (MSFT)** framework, which directly model $P(Y|do(X))$; (b): Challenges in directly applying the framework. *Left*: Downsampling and patching process for constructing multi-scale sequences. Patch tokens at different scales have varying resolution and schematics. *Right*: Directly applying self-attention over multi-scale embeddings leads to biased cross-scale attention due to misaligned time id.

ing scale-specific knowledge in the TSFM. Meanwhile, $X \rightarrow M$ reflects that the TSFM activates context-specific knowledge with the input data $X$.

$X \rightarrow Y \leftarrow M$. This link represents that the model utilizes the activated knowledge $M$ to generate predictions $Y$ based on the lookback context data $X$.

It is evident that scale $S$ is a confounder that induces spurious correlations between input context series (via $S \rightarrow X$) and activated knowledge of TSFM (via $S \rightarrow M$). The former captures the multi-scale properties of time series, while the latter corresponds to the multi-scale capabilities of TSFM. Scale $S$ ultimately affects the forecasting of the prediction horizon via the backdoor path $X \leftarrow S \rightarrow M \rightarrow Y$. Naive finetuned forecaster for $P(Y|X)$ overlooks the impact of this backdoor path, learning forecasting only at the original scale. This oversight would mistakenly associate non-causal but positively correlated input context to forecast horizon in the original scale, resulting in problematic forecasting. Further discussion can be found in Appendix C.

### 3.2 Causal Intervention via Backdoor Adjustment

Given this, we propose using $P(Y|do(X))$ as the new finetuned forecaster, which eliminates the confounding effect of $S$ and captures the true causal relationship from $X$ to $Y$. As the "physical" intervention is impossible, we apply the backdoor adjustment [27] to "virtually" realize $P(Y|do(X))$ by (1) blocking the link $S \rightarrow X$ and (2) S. As illustrated in Figure 2 (a, left), we have:

$$P(Y|do(X)) = \sum_s P(Y|X, S = s, M = g(X, s))P(s) \tag{2}$$

where $g$ is a function to activate scale-specific knowledge of input. Grounded in this causal formulation, we design the **MultiScale FineTuning** (**MSFT**) framework to instantiate the intervention-based forecasting process shown in Equation 2. As shown in the right panel of Figure 2(a), the framework stratifies the confounder $S$ by *down-sampling* the original time series into multiple scales. Each scale captures distinct statistical properties of the series and corresponds to a specific value $s \in S$.

Specifically, multi-scale series $\mathcal{S} = \{\mathbf{S}_0, \ldots, \mathbf{S}_K\}$ is generated through the process described in Section 2. Each scale series $\mathbf{S}_i$ is segmented into scale-specific patch tokens $\boldsymbol{x}_i \in \mathbb{R}^{N_i \times P}$, where $N_i$ is the number of patches for scale $i$. The scale-specific input embeddings are computed by $\boldsymbol{h}_i^0 = \text{InProject}(\boldsymbol{x}_i)$. Following the design of masked encoder [9], the embeddings falling within the forecast horizon are replaced with the learnable [mask] embedding. The input embeddings from all scales are concatenated into a multi-scale sequence, $\boldsymbol{h}_0 = \text{Concat}(\boldsymbol{h}_0^0, \boldsymbol{h}_1^0, \ldots, \boldsymbol{h}_K^0)$, which is then passed to the Transformer for processing.

### 3.3 Challenges

Although the framework of Figure 2(a) can be directly applied without initiation of $M = g(X, s)$, we argue that it leaves following challenges unaddressed. **First**, the token schematics and intra-scale dependencies vary significantly across scales. As shown in the left part of Figure 2(b), patch tokens at different scales exhibit distinct resolution and temporal schematics. When directly finetuning the

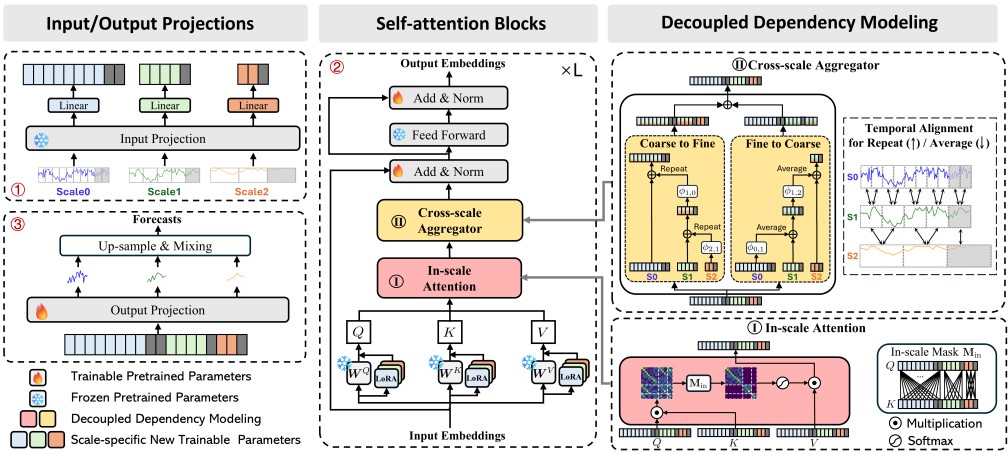

Figure 3: Complete design of MSFT based on the overall framework in Figure 2(a). ① Linear adapters are attached to the frozen input projection to learn scale-variant input embeddings. ② Self-attention layers incorperate scale-specific Lora and decoupled dependency modeling. ⓘ In-scale attention employs in-scale masking, ensuring tokens attend only to others within the same scale. ⓘⓘ Cross-scale aggregators progressively fuse tokens across scales in two directions, ensuring correct temporal alignment between tokens. ③ Output projection generates separate predictions for each scale, which are then mixed by up-sampling and learned weights.

input projection layer over all scales, each scale inherently tends to learn its own specific intra-token patterns, which can lead to interference across scales and suboptimal performance. Moreover, the resolution discrepancy induces scale-inequivalent inter-token correlation, requiring the attention mechanism to capture scale-specific dynamics rather than assuming uniform interaction patterns.

**Second**, standard self-attention introduces misleading cross-scale dependencies due to mismatched time (position) indices. Since time indices are independently generated within each scale, tokens with the same index at different scales (shaded in gray in Figure 2(b)) correspond to different temporal ranges. When self-attention is directly applied over over the concatenated multi-scale embedding sequence, attention scores across scales become biased: tokens attend more to others with the same time index, regardless of actual temporal relevance (see the right part of Figure 2(b)). This leads attention to capture spurious temporal correlations and attend to semantically irrelevant tokens.

**Finally**, the model generates distinct predictions at each scale, and effectively mixing multi-scale predictions remains a non-trivial challenge. Although cross-scale information is partially fused through attention, prior studies [42] have shown that explicitly combining multi-scale predictions improves forecasting performance. However, naively averaging predictions across scales fails to account for their semantic and temporal heterogeneity, potentially leading to suboptimal results.

## 4 Methodology

To address the aforementioned challenges, we propose **MSFT** to realize the high-level framework in Figure 2(a) as an effective multiscale finetuning strategy. Specifically, to **activate scale-specific knowledge**, we freeze the pretrained parameters and introduce scale-specific, parameter-efficient modules into the ① input projection and ② attention layers. To eliminate the cross-scale attention bias and correctly capture temporal correlations, we propose a **decoupled token dependency modeling** mechanism: ⓘ in-scale self-attention captures within-scale dependencies, while ⓘⓘ cross-scale aggregators explicitly fuse information across scales, ensuring correct temporal alignment between tokens. Finally, we apply **multi-scale mixing** to the ③ output projection, combining scale-specific predictions with learned weights. Figure 3 illustrates our MSFT method, with the detailed pseudo-code provided in Appendix B.1.

**Scale-specific Knowledge Activation.** To address the problem of scale-variant token resolution, instead of directly finetuning the unified input projection layer across all scales, we freeze the pretrained input projection and introduce a *scale-specific adapter* for each scale, implemented as a linear layer $\text{Linear}_i$. Now, the input embeddings of scale $i$ is computed as $\boldsymbol{h}_i^0 = \text{Linear}_i(\text{InProject}(\boldsymbol{x}_i))$.

Conditioned on the pretrained embeddings, these adapters independently learn specific representations at variant resolutions, effectively avoiding interference across scales.

Similarly, to enhance the attention mechanism's ability to capture scale-variant dynamics, we incorporate independent LoRA [13] modules for each scale. Specifically, we freeze the pretrained attention weight matrices, and the FFN block, and introduce a set of LoRA modules for each scale. Since both input embeddings and attention weights reflect scale-activated TSFM knowledge, this design serves as the implementation of $g$ in Equation 2, enabling the activation of scale-specific knowledge $M$.

**Decoupled Token Dependency Modeling.** To ensure attention blocks capture the multi-scale embedding sequence's correct dependencies, we decouple the token dependency modeling into two parts: within-scale and across-scale dependencies. Specifically, for tokens within the same scale, if they share the same resolution—dependencies, they can be directly learned via self-attention. Thus, we only apply an **in-scale attention mask** $\mathrm{M_{in}}$ to ensure that each token attends only to tokens from the same scale.

To aggregate the knowledge between tokens from different scales, we add a **cross-scale aggregator** after the attention operation. The aggregator consists of two branches, namely *coarse-to-fine* and *fine-to-coarse*, where temporal-aligned token-level information fusion is iteratively conducted between consecutive scales in two directions. First, since tokens at different scales correspond to varying resolutions, it is necessary to map embeddings to a shared space before fusion. To this end, following [29, 30], we adopt a linear mapping $\phi_{i,j}^l$ to project token embeddings from scale $i$ to the embedding space of scale $j$ in each layer $l$, where the mapped embeddings are defined as $\tilde{\boldsymbol{h}}_{i,j}^l = \phi_{i,j}^l(\boldsymbol{h}_i^l) = \boldsymbol{w}_{i,j}^l \boldsymbol{h}_i^l + \boldsymbol{b}_{i,j}^l$.

Based on this mapping, token embeddings from one scale are projected to the adjacent scale and then fused according to their temporal alignment. We define the cross-scale token-wise fusion for the *coarse-to-fine* (C2F) and *fine-to-coarse* (F2C) branches as follows:

$$\text{C2F:} \quad \boldsymbol{h}_{i-1}^l = \boldsymbol{h}_{i-1}^l + \text{Repeat}(\tilde{\boldsymbol{h}}_{i,i-1}^l), \qquad \text{for } i \in \{K, ..., 1\} \tag{3}$$

$$\text{F2C:} \quad \boldsymbol{h}_{i+1}^l = \boldsymbol{h}_{i+1}^l + \text{AvgPool}(\tilde{\boldsymbol{h}}_{i,i+1}^l), \quad \text{for } i \in \{0, ..., K-1\} \tag{4}$$

where $\text{Repeat}(\cdot)$ duplicates each coarse-scale token in $\tilde{\boldsymbol{h}}_{i,i-1}^l$ along the sequence dimension to match the finer-scale resolution, based on their temporal correspondence. Conversely, $\text{AvgPool}(\cdot)$ aggregates groups of fine-scale tokens in $\tilde{\boldsymbol{h}}_{i,i+1}^l$ by averaging them according to the downsampling factor, thereby aligning them to the coarser-scale resolution. Finally, the outputs from the two branches are combined by averaging their updated token embeddings. This decoupled two-stage design enables the model to capture temporal dependencies within each scale while effectively fusing complementary information across scales, leading to improved multi-scale temporal understanding.

**Multi-scale Mixing.** In the output projection, each scale independently predicts a forecasting horizon $\hat{\mathbf{Y}}_i$ based on its scale-specific tokens $\boldsymbol{h}_i^L$ from the final layer embedding $\boldsymbol{h}^L$. The training objective is formulated as a weighted summation of the scale-wise forecasting losses $\mathcal{L}_{\text{pred},i}$ (e.g., MSE or NLL). Since different scales may exhibit varying forecasting abilities and contribute differently to the final performance, we assign a learnable weight $w_i$ to each scale, corresponding to the prior $P(s)$ in Equation 2. The weights $w_i$ are obtained by applying a softmax function over a set of learnable parameters during training: $\mathcal{L}_{\text{pred}} = \sum_{i=0}^{K+1} w_i \mathcal{L}_{\text{pred},i}$. During inference, we upsample the forecasting results from each new scale to the original temporal resolution. The final prediction $\hat{\mathbf{Y}}$ is computed as the weighted sum of the upsampled forecasts, using the same learned weights $w_i$. This weighted mixing strategy can be seen as ensembling [26], which helps mitigate overfitting on the original scale. Additional implementation details for different TSFM architectures are provided in Appendix B.3.

# 5 Related Work

## 5.1 Time Series Foundation Model

We focus our discussion solely on transformer-based TSFMs for TSF. Such TSFMs can be broadly categorized according to the backbone architecture. Encoder-only models like Moirai [43], Moment [12] and UniTS [11] use masked reconstruction for pretraining. Decoder-only models, such

as TimesFM [8], Lag-Llama [32], Timer [20], and Time-MoE [35] are pretrained by next-token prediction in an auto-regressive manner. Chronos [2], an encoder-decoder model, quantizes scaled time series values into discrete tokens and adopts the training objective originally developed for NLP. Despite the advancement of the field, existing TSFM research predominantly emphasizes pretraining and zero-shot performance. Although some studies [2, 12, 8, 20] mention naive finetuning methods, these attempts are limited compared to the efforts devoted to pretraining and zero-shot evaluation. We include a more detailed discussion in Appendix A.

## 5.2 Multi-scale modeling in time series forecasting

Multi-scale modeling has garnered growing attention in the TSF community. Existing works mostly involves down-sampling, where coarser scales are derived from the original series using pooling or convolution. Models are then designed to capture multi-scale characteristics from these different views. Pyraformer [18] constructs a pyramidal graph of different scales and employs a pyramid attention mechanism to extract multi-resolution representations. MICN [40] processes different scales separately through multiple branches with distinct convolution kernels and subsequently merges the outputs. Inspired by hierarchical forecasting, Scaleformer [34] and GPHT [21] iteratively refine the outputs from coarser to finer scales. TimeMixer [42] and TimeMixer++ [41] decompose each scale into seasonal and trend components, then integrate these components across multiple scales.

# 6 Experiments

We evaluate our proposed finetuning method, MSFT, on two prevalent TSF tasks: long sequence forecasting (LSF) and probabilistic forecasting (PF). For LSF, we experiment with three TSFMs: MOIRAI, MOMENT and UNITS. For PF, we focus solely on MOIRAI, as it is the only model capable of probabilistic forecasting. Our evaluation includes comparisons with both deep learning-based methods and other finetuning approaches applied to TSFMs. Detailed model configurations and experimental setups are provided in the Appendix B.

Table 1: Long sequence forecasting results, which are averaged across prediction lengths $\{96, 192, 336, 720\}$. Each TSFM shows its zero-shot performance (highlighted in gray ) and results with different *finetuning methods*. The best finetuning results for each TSFM are highlighted in **bold**, while the global best results across all models are highlighted in **red**.

| Method | ETTm1 | | ETTm2 | | ETTh1 | | ETTh2 | | Electricity | | Weather | |
|---|---|---|---|---|---|---|---|---|---|---|---|---|
| | MSE | MAE | MSE | MAE | MSE | MAE | MSE | MAE | MSE | MAE | MSE | MAE |
| DLinear[2023] | 0.403 | 0.419 | 0.350 | 0.401 | 0.456 | 0.452 | 0.559 | 0.515 | 0.212 | 0.365 | 0.265 | 0.317 |
| PatchTST[2023] | 0.387 | 0.400 | 0.281 | 0.326 | 0.469 | 0.455 | 0.387 | 0.407 | 0.216 | 0.304 | 0.259 | 0.281 |
| iTransformer[2024a] | 0.407 | 0.410 | 0.288 | 0.332 | 0.454 | 0.448 | 0.383 | 0.407 | 0.178 | 0.270 | 0.258 | 0.278 |
| TimeMixer[2024] | 0.381 | 0.395 | 0.275 | 0.323 | 0.447 | 0.440 | 0.364 | 0.395 | 0.182 | 0.272 | 0.240 | 0.271 |
| SimpleTM [2025] | 0.381 | 0.396 | 0.275 | 0.322 | 0.422 | 0.428 | 0.353 | 0.391 | **0.166** | **0.260** | 0.243 | 0.271 |
| MOIRAI_Small | 0.448 | 0.409 | 0.300 | 0.341 | 0.416 | 0.428 | 0.355 | 0.381 | 0.233 | 0.320 | 0.268 | 0.279 |
| *+ Full finetuning* | 0.367 | 0.382 | 0.273 | 0.316 | 0.415 | 0.429 | 0.352 | 0.378 | 0.193 | 0.279 | 0.228 | 0.254 |
| *+ Linear probing* | 0.388 | 0.392 | 0.295 | 0.337 | 0.414 | 0.427 | 0.354 | 0.380 | 0.212 | 0.299 | 0.237 | 0.260 |
| *+ Prompt tuning* | 0.384 | 0.391 | 0.292 | 0.334 | 0.414 | 0.428 | 0.354 | 0.381 | 0.217 | 0.304 | 0.235 | 0.258 |
| *+ LoRA* | 0.370 | 0.383 | 0.272 | 0.314 | 0.414 | 0.427 | 0.354 | 0.380 | 0.192 | 0.279 | 0.225 | 0.252 |
| *+ AdaLoRA* | 0.381 | 0.386 | 0.273 | 0.319 | 0.414 | 0.427 | 0.354 | 0.380 | 0.191 | 0.279 | 0.226 | 0.252 |
| *+ MSFT* | **0.353** | **0.377** | **0.250** | **0.301** | **0.412** | **0.426** | **0.349** | **0.375** | **0.187** | **0.275** | **0.216** | **0.248** |
| MOIRAI_Base | 0.381 | 0.388 | 0.281 | 0.326 | 0.412 | 0.424 | 0.356 | 0.388 | 0.188 | 0.274 | 0.246 | 0.265 |
| *+ Full finetuning* | 0.368 | 0.371 | 0.258 | 0.307 | 0.409 | 0.424 | 0.357 | 0.384 | 0.173 | 0.263 | 0.232 | 0.258 |
| *+ Linear probing* | 0.388 | 0.387 | 0.277 | 0.319 | 0.409 | 0.424 | 0.356 | 0.387 | 0.182 | 0.269 | 0.229 | 0.253 |
| *+ Prompt tuning* | 0.378 | 0.386 | 0.280 | 0.325 | 0.412 | 0.423 | 0.360 | 0.387 | 0.183 | 0.271 | 0.230 | 0.255 |
| *+ LoRA* | 0.361 | 0.371 | 0.259 | 0.308 | 0.409 | 0.423 | 0.358 | 0.384 | 0.173 | 0.263 | 0.230 | 0.258 |
| *+ AdaLoRA* | 0.359 | 0.371 | 0.258 | 0.307 | 0.410 | 0.423 | 0.356 | 0.384 | 0.173 | 0.264 | 0.236 | 0.260 |
| *+ MSFT* | **0.332** | **0.369** | **0.247** | **0.305** | **0.407** | **0.422** | **0.352** | **0.383** | **0.169** | **0.260** | **0.213** | **0.245** |
| MOMENT | - | - | - | - | - | - | - | - | - | - | - | - |
| *+ Full finetuning* | 0.352 | 0.380 | 0.260 | 0.320 | 0.425 | 0.440 | 0.347 | 0.394 | 0.224 | 0.311 | 0.336 | 0.310 |
| *+ Linear probing* | 0.355 | 0.381 | 0.261 | 0.321 | 0.429 | 0.441 | 0.347 | 0.395 | 0.226 | 0.313 | 0.338 | 0.312 |
| *+ Prompt tuning* | 0.356 | 0.381 | 0.261 | 0.320 | 0.427 | 0.440 | 0.348 | 0.395 | 0.226 | 0.312 | 0.336 | 0.310 |
| *+ LoRA* | 0.356 | 0.381 | 0.260 | 0.320 | 0.425 | 0.439 | 0.347 | 0.395 | 0.225 | 0.312 | 0.335 | 0.309 |
| *+ AdaLoRA* | 0.355 | 0.381 | 0.259 | 0.319 | 0.426 | 0.440 | 0.347 | 0.394 | 0.224 | 0.311 | 0.336 | 0.311 |
| *+ MSFT* | **0.344** | **0.377** | **0.255** | **0.316** | **0.422** | **0.436** | **0.345** | **0.392** | **0.221** | **0.309** | **0.332** | **0.307** |
| UNITS | 0.713 | 0.553 | 0.321 | 0.355 | 0.527 | 0.491 | 0.406 | 0.418 | 0.432 | 0.488 | 0.291 | 0.313 |
| *+ Full finetuning* | 0.395 | 0.405 | 0.297 | 0.338 | 0.442 | 0.435 | 0.386 | 0.409 | 0.190 | 0.283 | 0.257 | 0.283 |
| *+ Linear probing* | 0.399 | 0.409 | 0.301 | 0.343 | 0.445 | 0.437 | 0.392 | 0.412 | 0.200 | 0.291 | 0.274 | 0.293 |
| *+ Prompt tuning* | 0.431 | 0.430 | 0.299 | 0.341 | 0.438 | 0.433 | 0.386 | **0.405** | 0.191 | 0.287 | 0.247 | 0.276 |
| *+ LoRA* | 0.393 | 0.405 | 0.296 | 0.338 | 0.437 | 0.434 | 0.384 | 0.407 | 0.188 | 0.282 | 0.250 | 0.279 |
| *+ MSFT* | **0.390** | **0.403** | **0.288** | **0.334** | **0.434** | **0.430** | **0.380** | **0.405** | **0.184** | **0.279** | **0.242** | **0.273** |

Table 2: Probabilistic forecasting results. The best finetuning results for each TSFM are highlighted in **bold**, while the global best results are highlighted in red. See Table 15 for full results.

| Method | Electricity | | Solar | | Weather | | Istanbul Traffic | | Turkey Power | |
|---|---|---|---|---|---|---|---|---|---|---|
| | CRPS | MSIS | CRPS | MSIS | CRPS | MSIS | CRPS | MSIS | CRPS | MSIS |
| DeepAR[2020] | 0.065 | 6.893 | 0.431 | 11.181 | 0.132 | 21.651 | 0.108 | 4.094 | 0.066 | 13.520 |
| TFT[2021] | 0.050 | 6.278 | 0.446 | 8.057 | 0.043 | 7.791 | 0.110 | 4.057 | 0.039 | 7.943 |
| PatchTST[2023] | 0.052 | 5.744 | 0.518 | 8.447 | 0.059 | 7.759 | 0.112 | 3.813 | 0.054 | 8.978 |
| TiDE[2023] | 0.048 | 5.672 | 0.420 | 13.754 | 0.054 | 8.095 | 0.110 | 4.752 | 0.046 | 8.579 |
| MOIRAI$_{Small}$ | 0.072 | 7.999 | 0.471 | 8.425 | 0.049 | 5.236 | 0.173 | 5.937 | 0.048 | 7.127 |
| + *Full finetuning* | 0.055 | 6.009 | 0.395 | 6.947 | 0.039 | 4.477 | 0.151 | 6.735 | 0.040 | 6.887 |
| + *Linear probing* | 0.062 | 6.438 | 0.369 | **5.865** | 0.049 | 4.785 | 0.154 | 4.645 | 0.047 | 6.912 |
| + *Prompt tuning* | 0.066 | 6.595 | 0.421 | 6.936 | 0.050 | 4.901 | 0.154 | 4.733 | 0.045 | 7.042 |
| + *LoRA* | 0.064 | 6.753 | 0.372 | 6.582 | 0.039 | 4.386 | 0.154 | 4.753 | 0.042 | 7.051 |
| + *AdaLoRA* | 0.064 | 6.892 | 0.366 | 8.015 | 0.040 | 4.496 | 0.152 | 4.670 | 0.041 | 7.127 |
| + *MSFT* | **0.047** | **5.327** | **0.353** | 7.706 | **0.036** | **4.178** | **0.141** | **4.447** | **0.038** | **6.810** |
| MOIRAI$_{Base}$ | 0.055 | 6.172 | 0.419 | 7.011 | 0.041 | 5.136 | 0.116 | 4.461 | 0.040 | 6.766 |
| + *Full finetuning* | 0.049 | 5.414 | 0.188 | 4.292 | 0.038 | 5.282 | 0.120 | 7.272 | 0.036 | 6.712 |
| + *Linear probing* | 0.055 | 5.951 | 0.379 | 5.645 | 0.039 | 4.544 | 0.104 | 3.736 | 0.042 | 7.259 |
| + *Prompt tuning* | 0.054 | 6.024 | 0.412 | 6.885 | 0.040 | 5.274 | 0.105 | 3.987 | 0.040 | 6.698 |
| + *LoRA* | 0.051 | 5.651 | 0.382 | 6.745 | 0.037 | 4.904 | 0.113 | 4.752 | 0.036 | 6.744 |
| + *AdaLoRA* | 0.054 | 5.937 | 0.383 | 8.825 | 0.038 | 4.802 | 0.110 | 3.895 | 0.037 | 6.762 |
| + *MSFT* | **0.046** | **5.199** | **0.142** | **3.464** | **0.035** | **4.603** | **0.098** | **3.685** | **0.034** | **6.419** |

## 6.1 Long Sequence Forecasting

**Setup.** We conduct our experiments on a subset of the widely-used long sequence forecasting benchmark [45]. This subset is identical to the one used in Moirai [43] for LSF experiments and is not included in the pretraining data of TSFMs. Each dataset involves predictions at four different lengths, with the model is finetuned separately for each prediction length. We evaluate the performance using Mean Squared Error (MSE) and Mean Absolute Error (MAE).

**Results.** As shown in Table 1, MSFT consistently enhances the forecasting performance of TSFMs. Across all models, MSFT outperforms other finetuning methods that use only the original scale, consistently delivering the best finetuned results. For MOIRAI$_{Small}$ and MOIRAI$_{Base}$, MSFT further improves their forecasting accuracy over their solid zero-shot performance, achieving competitive results across all datasets, with 10 out of 12 metrics showing the best performance. Notably, MSFT substantially improves MOIRAI's finetuned performance on minutely-level datasets. Compared to full finetuning, it achieves 6.8% lower MSE in ETTm1, 6.3% lower MSE in ETTm2 and 6.7% lower MSE in Weather. In contrast, the improvement brought by MSFT on hourly datasets are relatively smaller compared to minute-level datasets. This discrepancy can be explained by the richer multi-scale patterns present in minute-level data, which MSFT can effectively leverage. For MOMENT, the improvements brought by MSFT are generally less pronounced compared to MOIRAI and UNITS. This can be attributed its pretraining with fixed context lengths, which limits their ability to extract information from new scales of varying lengths. Despite these differences, MSFT exhibit superior finetuned performance across diverse models and datasets, demonstrating its generalizability.

## 6.2 Probabilistic Forecasting

**Setup.** We evaluate on six datasets spanning various domains, using the rolling evaluation setup described in Moirai [43]. The test set comprises the final time steps, segmented into multiple non-overlapping evaluation windows. The length of the prediction window and the number of rolling evaluations are tailored for each dataset based on its frequency (see Table 5 for details). For performance evaluation, we report the Continuous Ranked Probability Score (CRPS) and Mean Scaled Interval Score (MSIS) metrics.

**Results.** Experimental results in Table 2 demonstrate that MSFT consistently delivers superior performance across all datasets. Building upon the strong zero-shot performance, MOIRAI$_{Base}$ achieves the best results for nearly all the datasets. MSFT provides consistent improvements over other finetuning methods, achieving an additional 24.4 % CPRS relative reduction in Solar and 18.3 % CPRS relative reduction in Istanbul Traffic compared to full finetuning. A similar trend is also observed in the small model, demonstrating that our multi-scale modeling method can effectively enhance the fine-tuned performance of probabilistic forecasting.

Table 3: Ablation study on three LSF datasets using MOIRAI$_{Small}$.

| | InProject | | Attention | | In-scale Mask | X-scale Aggre. | | Mixing | | ETTm1 | | ETTm2 | | Weather | | Avg Diff | |
|---|---|---|---|---|---|---|---|---|---|---|---|---|---|---|---|---|---|
| | Scale | Shared | Scale | Shared | | C2F | F2C | Avg. | Weighted | MSE | MAE | MSE | MAE | MSE | MAE | MSE | MAE |
| ① | | | ✓ | | ✓ | ✓ | ✓ | | ✓ | 0.362 | 0.380 | 0.253 | 0.305 | 0.219 | 0.252 | 0.005 | 0.003 |
| ② | | ✓ | ✓ | | ✓ | ✓ | ✓ | | ✓ | 0.360 | 0.379 | 0.252 | 0.304 | 0.218 | 0.249 | 0.003 | 0.002 |
| ③ | ✓ | | | | ✓ | ✓ | ✓ | | ✓ | 0.374 | 0.385 | 0.256 | 0.308 | 0.224 | 0.256 | 0.011 | 0.007 |
| ④ | ✓ | | | ✓ | ✓ | ✓ | ✓ | | ✓ | 0.361 | 0.382 | 0.254 | 0.306 | 0.222 | 0.254 | 0.006 | 0.005 |
| ⑤ | ✓ | | ✓ | | ✓ | | | | ✓ | 0.371 | 0.384 | 0.256 | 0.307 | 0.223 | 0.255 | 0.010 | 0.006 |
| ⑥ | ✓ | | ✓ | | ✓ | | ✓ | | ✓ | 0.363 | 0.382 | 0.254 | 0.304 | 0.220 | 0.252 | 0.006 | 0.004 |
| ⑦ | ✓ | | ✓ | | ✓ | ✓ | | | ✓ | 0.357 | 0.379 | 0.252 | 0.304 | 0.218 | 0.251 | 0.002 | 0.002 |
| ⑧ | ✓ | | ✓ | | | | | | ✓ | 0.360 | 0.380 | 0.253 | 0.303 | 0.220 | 0.253 | 0.004 | 0.003 |
| ⑨ | ✓ | | ✓ | | ✓ | ✓ | ✓ | | | 0.359 | 0.379 | 0.269 | 0.313 | 0.226 | 0.252 | 0.011 | 0.006 |
| ⑩ | ✓ | | ✓ | | ✓ | ✓ | ✓ | ✓ | | 0.384 | 0.388 | 0.255 | 0.311 | 0.219 | 0.252 | 0.012 | 0.008 |
| **MSFT** | ✓ | | ✓ | | ✓ | ✓ | ✓ | | ✓ | **0.354** | **0.378** | **0.250** | **0.301** | **0.216** | **0.248** | - | - |

## 6.3 Model Analysis

To fully understand MSFT, we conduct model analysis using the MOIRAI$_{Small}$ model on three LSF datasets, selected for its strong zero-shot performance and relatively low training cost. Due to page limits, we present the analysis of down-sampling approaches, down-sampling factors, detailed attention analysis, and visualizations in the Appendix D. We also discuss the potential application of MSFT to decoder-based structures and its limitation in Appendix E.

**Ablation Study.** To ensure statistical robustness, we report mean results over three runs in Table 3, with standard deviations provided in Table 11. Ablations ① to ④ examine the effectiveness of scale-specific knowledge activation. For both input projection and attention, either freezing (①, ③) or finetuning shared weights (②, ④) yields inferior performance to using scale-specific modules, with freezing causing larger performance drops. Among the two, attention has a greater impact than input projection, highlighting its critical role in capturing temporal dependencies.

Ablations ⑤ to ⑧ evaluate the effect of each component in decoupled dependency modeling. In ⑤, we remove cross-scale aggregators and only retain in-scale attention masking. Without cross-scale modeling, the performance suffers a significant decline. In ⑥ and ⑦, we ablate the coarse-to-fine and fine-to-coarse branches, respectively. Both cases lead to performance drops, with the coarse-to-fine branch showing a stronger impact. In ⑧, we completely remove decoupled dependency modeling, capturing dependency directly via attention on the concatenated multi-scale sequence. This approach leads to misaligned cross-scale interactions and further degrades performance.

Finally, we assess the impact of multi-scale mixing. In ⑨, we disable prediction mixing, only using the original scale for prediction. In ⑩, we aggregate the multi-scale predictions by averaging. Both approaches result in lower performance compared to our full model.

**Effect of Number of New Scales.** As shown in Figure 4, increasing the number of new scales $K$ initially reduces errors. However, beyond a certain point, performance plateaus or declines, likely due to overly coarse predictions with few tokens disrupting multi-scale modeling. Our results indicate that setting $K$ to 2 or 3 achieves the best balance.

**Attention Analysis.** Figure 5 shows the attention score heatmaps of three attention strategies. In (a), direct attention (Ablation ⑧) exhibits spurious temporal dependencies, with attention scores biased toward tokens sharing the same time indices. In (b), we align time indices during attention, ensuring that cross-scale tokens corresponding to the same temporal region share identical time indices. While this approach produces "correct" attention patterns, it is limited to RoPE and performs worse than our method (see Appendix D for details). In (c), our in-scale masking strategy eliminates misleading cross-scale attention, focusing on accurate within-scale dependency modeling.

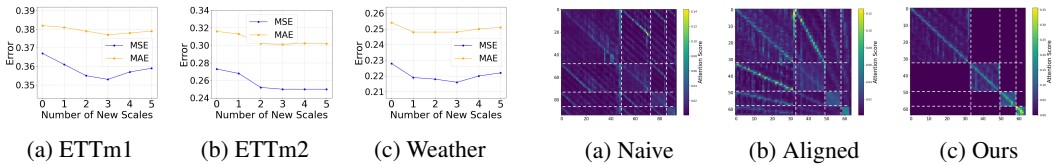

| (a) ETTm1 | (b) ETTm2 | (c) Weather | (a) Naive | (b) Aligned | (c) Ours |
|---|---|---|---|---|---|

Figure 4: LSF accuracy w.r.t. number of scales     Figure 5: Attention heatmaps of various methods

# 7 Conclusion

We introduce MSFT, a multi-scale finetuning strategy for encoder-based TSFMs. From a causal view, we highlight the limitations of naive finetuning and propose to use multi-scale modeling as backdoor adjustment to mitigate the confounding effect of scale. By using concatenated multi-scale sequence as input, applying simple scale-specific model modifications, and employing decoupled dependency modeling, our method effectively aggregates multi-scale information and improves the forecasting performance. Our experiments show that MSFT not only significantly enhances the performance of the original foundation models but also surpasses other state-of-the-art models trained from scratch.

## Acknowledgments and Disclosure of Funding

This research is part of the programme DesCartes and is supported by the National Research Foundation, Prime Minister's Office, Singapore under its Campus for Research Excellence and Technological Enterprise (CREATE) programme. Zhongzheng Qiao is affiliated with Energy Research Institute @ NTU, Interdisciplinary Graduate Programme, Nanyang Technological University, Singapore.

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

# A  Detailed Related Works

## A.1  Time Series Foundation Model

In this section, we further discuss related works involved TSFM finetuning. Some TSFMs conduct basic experiments in their original papers to show the effect of naive finetuning. For instance, Moment [12] applies linear probing and reports the corresponding results in their main experiments. Chronos [2] and Timer [20] apply full finetuning, with improvements over zero-shot perofrmance on their respective benchmarks. TimesFM [8] tunes the input and output residual blocks on ETT datasets.

Notably, some recent studies explore specialized finetuning methods for TSFMs. UniTS [11] introduces a prompt tuning strategy, and [6] employs in-context tuning to TimesFM. However, both methods are tailored to their own model architectures and rely on specialized pretraining designs, which limits their generalizability and plug-and-play applicability. [3] inserts adapters on Moment to adapt this univaraite TSFM for multivariate probabilistic forecasting. However, this approach is restricted to this specific application, rather than serving as a general finetuning method applicable to various models or forecasting tasks. This gap underscores the need for more general, effective, and modular finetuning strategies for TSFMs.

## A.2  Multi-scale modeling

The term multi-scale modeling has been used inconsistently in prior time series works. In our paper, it specifically refers to constructing multiple downsampled versions of the same time series and jointly leveraging them through cross-scale aggregation during prediction. This enables the model to integrate temporal information from both coarse and fine resolutions within a single forward pass.

Several related terminologies exist in the literature. TTM[10] introduces *multi-resolution pretraining* by downsampling high-frequency datasets into lower-frequency versions, which serves as a form of data augmentation during pretraining. However, each time series sample is still processed in a single scale/resolution during its prediction process. Thus, TTM does not perform multi-scale modeling as defined in our work. Another similar strategy is *multi-patch-size modeling*, which applies specifically to patch-based TSTs. Here, multiple patch sizes are used to segment the time series into tokens in different resolutions. Pathformer [5] applies layer-by-layer routing to select patch sizes and aggregate the outputs from multiple scales. MTST [49] proposes a multi-branch architecture for modeling diverse temporal patterns at different resolutions. ElasTST [47] leverages a shared transformer backbone with tunable RoPE for multi-scale patch assembly. Moirai [43] adopts multiple patch-size projection layers, yet it does not downsample inputs into multiple temporal scales. In summary, this line of methods requires the model to be compatible with multiple patch sizes, which is not feasible for most TSFMs. Under this clarified definition, our work is the first to introduce multi-scale modeling in TSFM finetuning.

# B  Implementation Details

## B.1  Pseudo-code of MSFT

For clarity, we provide the Pytorch-like pseudo codes of MSFT in Algorithm 1 and Algorithm 2, , illustrating the overall training pipeline and the MSFT attention block described in Section 4. Importantly, the method does not construct a new model from scratch, but enhances the pretrained TSFM through additional plug-in modules.

## B.2  Dataset details

For long sequence forecasting (LSF), we conduct experiments on six well-established datasets, including the ETT datasets (ETTh1, ETTh2, ETTm1, ETTm2) [51], Weather [45], and Electricity [45]. We note that these datasets are not included in the pretraining datasets of the TSFMs we evaluated. The key properties of these LSF datasets are detailed in Table 4.

**Algorithm 1** Overall Training Pipeline for MSFT: PyTorch-like Pseudo-code

```
class MSFTPipeline(nn.Module):
    def __init__(self, pretrained_model, K, s=2, P=16):
        self.K = K # number of new scales
        self.s = s # downsample factor
        self.P = P # patch size

        # frozen input projection, scale-specific linear layers
        self.in_proj = pretrained_model.in_proj
        self.linears = [Linear_i() for i in range(K+1)]

        # encoder layers with MSFT blocks
        self.encoder = nn.ModuleList([
            AttnBlock_with_MSFT(block, K=K, s=s)
            for block in pretrained_model.encoder_layers
        ])

        # frozen output projection
        self.out_proj = pretrained_model.out_proj

    def forward(self, X, Y):
        # Step 1: Multi-Scale Generation
        S = []
        for i in range(self.K+1):
            X_i = AvgPool(X, window_size=self.s**i) # pre-pad if needed
            Y_i = AvgPool(Y, window_size=self.s**i) # post-pad if needed
            S.append((X_i, Y_i))

        # Step 2: Patching & Projection
        H_0 = []
        for i, (X_i, Y_i) in enumerate(S):
            x_i = Patching((X_i, Y_i), patch_size=self.P)
            h_i = self.linears[i](self.in_proj(x_i)) # frozen InProject
            h_i = Masking(h_i) # mask prediction tokens
            H_0.append(h_i)
        h = Concat(H_0)
        scale_index = GetScaleIndex(H_0)

        # Step 3: Multi-Scale Attention Encoding
        for l in range(self.L):
            h = self.encoder[l](h, scale_index)

        # Step 4: Output & Loss
        H_L = Split(h, scale_index) # recover [h_0^L, ..., h_K^L]
        losses, preds = [], []
        for i, (_, Y_i) in enumerate(S):
            Y_hat_i = self.out_proj(H_L[i])
            L_i = Loss(Y_i, Y_hat_i)
            losses.append(w_i * L_i) # weighted by learnable w_i

            # upsample prediction back to original scale
            Y_hat_up = Upsample(Y_hat_i, scale=self.s**i)
            preds.append(w_i * Y_hat_up)

        L_total = sum(losses)
        Y_hat = sum(preds)
        return L_total, Y_hat
```

Following Moirai [43], we use 5 out-of-distribution datasets for probabilistic forecasting: Electricity [38], Solar-Power [16], Jena Weather, Istanbul Traffic[2], and Turkey Power[3]. Detailed descriptions of these datasets are provided in Table 5.

## B.3 Encoder-based TSFMs

We describe the architectural details and training objectives of each encoder-based TSFM used in our experiments. Table 6 summarizes the fundamental details of the models based on their origin setup.

---

[2]https://www.kaggle.com/datasets/leonardo00/istanbul-traffic-index
[3]https://www.kaggle.com/datasets/dharanikra/electrical-power-demand-in-turkey

**Algorithm 2** Multi-Scale Attention Block (AttnBlock_with_MSFT): PyTorch-like Pseudo-code

```python
class AttnBlock_with_MSFT(PretrainedAttnBlock):
    def __init__(self, K, s=2):
        super.__init__()
        self.K = K # number of new scales
        self.s = s # downsample factor

        # LoRA adapters for each scale
        self.W_Q = [LoRA(self.W_Q) for _ in range(K+1)]
        self.W_K = [LoRA(self.W_K) for _ in range(K+1)]
        self.W_V = [LoRA(self.W_V) for _ in range(K+1)]

        # cross-scale projections
        self.F2CMap = [Linear() for i in range(K)] # fine -> coarse
        self.C2FMap = [Linear() for i in range(K)] # coarse -> fine

    def forward(self, h_in, scale_index):
        # Step 1: Split input into scale-wise representation
        H_in = [h_in[..., idx, :] for idx in scale_index]

        # Step 2: Scale-specific Attention with LoRA
        Q, K, V = [], [], []
        for i in range(self.K+1):
            Q.append(W_Q[i](H_in[i]))
            K.append(W_K[i](H_in[i]))
            V.append(W_V[i](H_in[i]))
        Q, K, V = Concat(Q), Concat(K), Concat(V)

        # Step 3: In-scale masked attention
        h_attn = ScaledDotProductAttention(Q, K, V, mask=M_in)

        # Step 4: Cross-scale Aggregation
        H_attn = [h_attn[..., idx, :] for idx in scale_index]
        # (a) Coarse-to-Fine (C2F)
        H_c2f = H_attn.copy()
        for i in range(self.K, 0, -1):
            h_proj = self.C2FMap[i-1](H_attn[i])
            H_c2f[i-1] += Repeat(h_proj, repeat_factor=self.s)
        # (b) Fine-to-Coarse (F2C)
        H_f2c = H_attn.copy()
        for i in range(self.K):
            h_proj = self.F2CMap[i](H_attn[i])
            H_f2c[i+1] += AvgPool(h_proj, pool_size=self.s)
        # (c) Merge outputs from two branches
        H_out = []
        for i in range(self.K+1):
            H_out.append(0.5 * (H_c2f[i] + H_f2c[i]))

        # Step 5: Re-concatenate
        h_out = Concat(H_out)

        # W_o & Add & Norm & FeedForward omitted for brevity
        return h_out
```

**Moirai**    Moirai [43] is one of the pioneering TSFMs for universal time series forecasting based on a masked encoder architecture. It segments single-dimensional time series (a variate) into patch tokens and can be extended to multivariate setup by flattening multiple variate into a single sequence. Moirai employs multi patch size projection layers in both input and output projections, allowing it to effectively handle data with varying frequencies. In the attention blocks, it encodes the temporal position of tokens using Rotary Positional Encoding (RoPE) [37], and encodes simple variate correlation by using binary attention biases to indicate whether two tokens belong to the same variate or not. The model produces distribution parameters for a mixture distribution over the predictive horizon. The training objective is to minimize the negative log-likelihood (NLL). During inference, predictions of horizon are obtained by sampling from the predictive distribution. Point forecasts can be derived by taking the median from the samples. In our experiments, we use the univariate mode of Moirai, encoding different scales using distinct variate indices.

**Moment**    Moment [12] is a suite of open-source foundation models designed for versatile time-series analysis tasks. Moment follows channel independence assumption and leverages a T5[31]

Table 4: Summary of datasets used in the long sequence forecasting evaluation.

| Task | Dataset | Variate | Dataset Size | Predict Length | Frequency | Information |
|---|---|---|---|---|---|---|
| Long Sequence Forecasting | ETTh1 | 7 | 17420 | {96, 192, 336, 720} | Hourly | Temperature |
| | ETTh2 | 7 | 17420 | {96, 192, 336, 720} | Hourly | Temperature |
| | ETTm1 | 7 | 69680 | {96, 192, 336, 720} | 15 min | Temperature |
| | ETTm2 | 7 | 69680 | {96, 192, 336, 720} | 15 min | Temperature |
| | Electricity | 321 | 26304 | {96, 192, 336, 720} | Hourly | Electricity |
| | Weather | 21 | 52696 | {96, 192, 336, 720} | 10 min | Weather |

Table 5: Summary of datasets used in the probabilistic forecasting evaluation setting.

| Task | Dataset | Variate | Dataset Size | Predict Length | Rolling Evaluation | Frequency | Information |
|---|---|---|---|---|---|---|---|
| Probabilistic Forecasting | Electricity | 321 | 26304 | 24 | 7 | H | Energy |
| | Solar | 137 | 8760 | 24 | 7 | H | Energy |
| | Weather | 21 | 52696 | 144 | 7 | 10T | Climate |
| | Istanbul Traffic | 3 | 14244 | 24 | 7 | H | Transport |
| | Turkey Power | 18 | 26304 | 24 | 7 | H | Energy |

encoder architecture enhanced with sinusoidal positional encoding to effectively capture temporal dependencies within time series. Distinctively, during the forecasting fine-tuning phase, MOMENT utilizes the entire context series as input to directly get prediction results, diverging from traditional masked reconstruction methods commonly employed in pretraining. The model's forecasting head comprises a flatten operation followed by a linear layer, and it is trained using the MSE loss function. Due to the computational resource constraints associated with finetuning and the large scale of the models, we employ Moment (Small) for our experiments.

**UNITS** UNITS is originally designed for multi-tasks learning with specific task prompts. The transformer encoder is composed of multiple UNITS Blocks and ultimately processed through the GEN Tower to generate the final predictions. Specifically, within each UNITS Block, the data sequentially passes through Time Self-attention, Variable Self-attention, and Dynamic FFN. Each of these modules is followed by a Gate Module, which enhances the model's generalization capability in multi-task learning by dynamically scaling the feature vectors. Time Self-attention and Variable Self-attention compute attention scores along the time and variable dimensions, respectively, while the Dynamic FFN dynamically adjusts the shape of the weight matrix through bilinear interpolation to match the lengths of the input and output. The GEN Tower is designed to accommodate varying input lengths for different tasks and to ultimately generate the output sequence. The model applies learnable additive position encoding. For forecasting task, the training objective is MSE loss.

### B.4 Finetuning baselines

**Full Finetune and Linear Probe** Full finetuning involves updating all parameters of the pretrained model. We observe that using a small learning rate is crucial for stability and performance. In contrast, linear probing only updates the output head while keeping the backbone frozen; a larger learning rate is generally more effective in this case.

**LoRA and AdaLoRA** LoRA [13] introduces trainable rank-decomposition matrices into the attention layers, enabling parameter-efficient finetuning by injecting updates into a low-rank subspace. AdaLoRA [48] extends LoRA by dynamically allocating the rank during training based on parameter importance, improving adaptation under a parameter budget. For Moirai and Moment, we directly adopt the PEFT library [22] for both LoRA and AdaLoRA. We apply LoRA and AdaLoRA to the query, key, and value projection layers. In addition to the LoRA modules, we also allow the output prediction head to be trainable. The LoRA configuration follows standard settings with rank $r = 16$ and scaling factor $\alpha = 32$. For AdaLoRA, we use the default configuration provided by the PEFT library. Since the original attention implementation in the UNITS codebase uses a large shared weight

Table 6: Summary of encoder-based time series foundation models.

| Feature | Moirai | Moment | UNITS |
|---|---|---|---|
| Citation | Woo et al., 2024 | Goswami et al., 2024 | Gao et al., 2024 |
| Base Architecture | Naive Encoder | T5 Encoder | Modified Encoder |
| Params | *Small*: 14M , *Base*: 91M | 40M | 3.4M |
| Open Source | ✓ | ✓ | ✓ |
| Evaluation Tasks | Point Forecasting, Probabilistic Forecasting | Point Forecasting, Classification, Anomaly detection, Imputation | Point Forecasting, Classification, Anomaly detection, Imputation |
| Layer | *Small*: 6 , *Base*: 12 | 8 | 3 |
| $d_{model}$ | *Small*: 384, *Base*: 768 | 512 | 128 |
| Patch Size | [8, 16, 32, 64, 128] | 8 | 16 |
| Context Length | 1000-5000 | 512 | 96 |
| Position Embedding | RoPE [37] | Sinusoidal | Learnable Additive PE |

for query, key, and value, applying LoRA or AdaLoRA from PEFT is not feasible. Therefore, we implement a custom LoRA for it and do not conduct AdaLoRA experiments on UNITS.

**Prompt Finetuning** For Moirai and Moment, we implement prompt fine-tuning by introducing trainable soft prompt embeddings, which are prepended to the input tokens in the embedding space. We avoid inserting them into the patch token space, as doing so can interfere with the statistical computation of RevIN [14] and offers less expressive capacity compared to the high-dimensional embedding space. During inference, we discard the prompt embeddings from the encoder output and use only the time series embeddings as final representation for prediction. Similarly, only the output head and the prompt embeddings are finetuned, while all other parameters remain frozen. Prompt length is set to 2 by default. For UNITS, we directly use its original prompt tuning implementation.

### B.5 Metric details

For long sequence forecasting, we follow the standard protocols to use mean square error (MSE) and mean absolute error for evaluation. For probabilistic forecasting, we include Continuous Ranked Probability Scoremean (CRPS), Mean Scaled Interval Score (MSIS), absolute percentage error (MAPE), symmetric mean absolute percentage error (sMAPE), mean absolute scaled error (MASE), normalized deviation (ND), and normalized root mean squared error (NRMSE) as metrics. The definitions and calculations of probabilistic forecasting metrics are as follows. Note that the notations used here are independent of those in the main text.

**Continuous Ranked Probability Score** Given a predicted distribution with c.d.f. $F$ and ground truth $\mathbf{Y}$, the CRPS is defined as:

$$\text{CRPS} = \int_0^1 2\Lambda_\alpha(F^{-1}(\alpha), \mathbf{Y})d\alpha$$
$$\Lambda_\alpha(q, \mathbf{Y}) = (\alpha - \mathbf{1}_{\mathbf{Y}<q})(\mathbf{Y} - q),$$

where $\Lambda_\alpha$ is the $\alpha$-quantile loss, also known as the pinball loss at quantile level $\alpha$.

In practice, the CRPS is intractable or computationally expensive to compute, and we also want to compute a normalized metric, thus we compute a normalized discrete approximation, the mean weighted sum quantile loss, defined as the average of $K$ quantiles:

$$\text{CRPS} \approx \frac{1}{K}\sum_{k=1}^{K}\text{wQL}[\alpha_k]$$
$$\text{wQL}[\alpha] = 2\frac{\sum_i \Lambda_\alpha(\hat{q}_i(\alpha), \mathbf{Y}_i)}{\sum_i |\mathbf{Y}_i|},$$

where $\mathbf{Y}_i$ is the ground truth at at time step $i$ and $\hat{q}_t(\alpha)$ is the predicted $\alpha$-quantile at time step $i$. We take $K = 9, \alpha_1 = 0.1, \alpha_2 = 0.2, \ldots, \alpha_9 = 0.9$ in practice.

**Mean Scaled Interval Score**    The MSIS is a metric to evaluate uncertainty around point forecasts. Given an upper bound prediction, $U_i$, and lower bound prediction $L_i$, the MSIS is defined as:

$$\text{MSIS} = \frac{1}{H \cdot \left( \frac{1}{n-m} \sum_{i=m+1}^{n} |\mathbf{Y}_i - \mathbf{Y}_{i-m}| \right)} \cdot \left[ \sum_{i=1}^{H} (U_i - L_i) \right.$$
$$\left. + \frac{2}{a}(L_i - \mathbf{Y}_i)\mathbb{1}_{\{\mathbf{Y}_i < L_i\}} + \frac{2}{a}(\mathbf{Y}_i - U_i)\mathbb{1}_{\{\mathbf{Y}_i > U_i\}} \right]$$

where $a = 0.05$ is the significance level for a 95% prediction interval, over a forecast horizon of length $H$, and $m$ is the seasonal factor.

**symmetric Mean Absolute Percentage Error**    The sMAPE is a accuracy measure based on percentage errors, treating over- and under-predictions symmetrically, commonly used in forecasting.

$$\text{SMAPE} = \frac{200}{H} \sum_{i=1}^{H} \frac{|\mathbf{Y}_i - \widehat{\mathbf{Y}}_i|}{|\mathbf{Y}_i| + |\widehat{\mathbf{Y}}_i|},$$

**Mean Absolute Scaled Error**    The MASE is a metric for forecasting accuracy, scaling errors by the in-sample mean absolute error of a naive forecast, ensuring interpretability and comparability.

$$\text{MASE} = \frac{1}{H} \sum_{i=1}^{H} \frac{|\mathbf{Y}_i - \widehat{\mathbf{Y}}_i|}{\frac{1}{H-s} \sum_{j=s+1}^{H} |\mathbf{Y}_j - \mathbf{Y}_{j-s}|},$$

where $s$ is the periodicity of the data. $\mathbf{Y}, \widehat{\mathbf{Y}} \in \mathbb{R}^{H \times D}$ are the ground truth and prediction results of the future with $H$ time pints and $D$ dimensions. $\mathbf{Y}_i$ means the $i$-th future time point.

**Normalized Deviation**    The ND measures prediction accuracy by standardizing deviations between predicted and actual values, aiding model evaluation and optimization.

$$\text{ND} = \frac{1}{H} \sum_{i=1}^{H} \left| \frac{\mathbf{Y}_i - \hat{\mathbf{Y}}_i}{\mathbf{Y}_i} \right| \times 100\%,$$

**Normalized Root Mean Squared Error**    The NRMSE quantifies prediction error, enables model comparison, aids optimization, and provides interpretable results in time series forecasting.

$$\text{NRMSE} = \frac{\sqrt{\frac{1}{H} \sum_{i=1}^{H} \left( \mathbf{Y}_i - \hat{\mathbf{Y}}_i \right)^2}}{\max(\mathbf{Y}) - \min(\mathbf{Y})}.$$

### B.6    Experiment Details

**Dataset Construction**    Unlike pretraining in Moirai, where samples are randomly cropped from time series of varying lengths, we create the training, validation, and test datasets by cropping time series windows with fixed sequence lengths. Given the context and prediction lengths, samples are segmented using a sliding window, where the window size is $C + H$. The train-val-test split follows the default LSF setup. Data are normalized for LSF but not for PF.

**Training Setup**    Since there is no official fine-tuning implementation for Moirai, we configure the training setup as follows. We use the AdamW optimizer with weight decay=0.1, $\beta_1 = 0.9$, and $\beta_2 = 0.98$ for optimization. Specifically, unlike pretraining, which uses a learning rate of 1e-3, we find that finetuning requires a much smaller learning rate. Based on validation performance, we select a learning rate of either 5e-6 or 5e-7 for finetuning our models. The batch size is set to 512 by default for experiments using MOIRAI$_{\text{Small}}$, and reduced to 256 on MOIRAI$_{\text{Base}}$ if GPU memory reaches its limit. We adopt a constant learning rate scheduling, and early stopping is employed to monitor training. The context lengths are used directly from the values in the original Moirai models, which are tuned from a range of [1000, 2000, 3000, 4000, 5000]. The patch sizes are also taken from their provided values, which are selected based on data frequency. Since all samples have the same sequence length, sequence packing is not used during training. For Moment and UNITS, we directly follow their provided their original finetuning configurations for experiments, with the learning rate selected from 5e-5, 5e-6, or 5e-7.

**Evaluation Setup**    For Moirai, the evaluation is based on the GluonTS Library [1]. Predictions are sampled 100 times from the learned predicitive distributions, and evaluation metrics are computed over those samples. For Moment and UNITS, the LSF metrics are directly computed based on the output predicted series.

**Computational environment**    Our experiments are conducted on a server equipped with an AMD EPYC 7763 CPU (64 cores, 128 threads) and four NVIDIA A40 GPUs, each with 40 GB of memory.

## C    Causal Analysis

### C.1    Causal Modeling Motivation

Here we elaborate the motivation of our causal modeling in Figure 2. A time series can be viewed as a discretized sequence of sampled observations derived from an underlying continuous process. Under this perspective, the observed input window $X$ corresponds to a discrete observation of the latent process during a context period. The variable $X$ arises from two factors: the latent continuous process $I$ (unobserved) and the scale parameter $S$, which governs the sampling resolution. The scale $S$ determines how densely the latent process is sampled, thereby shaping both the temporal granularity and the length of the observed sequence $X$. While our formulation omits the latent process $I$ for tractability, the edge $S \rightarrow X$ in our causal graph reflects this observation mechanism. Importantly, scale influences the form of the observed input series but not the latent process itself.

Formally, we treat both $S$ and $X$ as random variables. The scale $S$ is a discrete variable that selects the resolution level for downsampling the input context. It takes values from a finite index set $\mathcal{S} = \{s_0, s_1, \ldots, s_K\}$, where each $s_k$ corresponds to a specific downsampling factor or temporal resolution. The observed input $X$ is then a random variable whose sequence length depends on the selected scale $S$. Thus, $X \in \mathcal{X}$, where $\mathcal{X} = \bigcup_{s \in \mathcal{S}} \mathbb{R}^{L_s}$ and $L_s$ denotes the input length corresponding to scale $s$. This formalization clarifies the role of scale in shaping observed input time series, consistent with the causal edge $S \rightarrow X$ in our proposed graph.

### C.2    Empirical Validation of the Causal Graph

To empirically validate the proposed causal graph, we perform causal structure learning and partial correlation analysis on the ETTm1 dataset across multiple scales. Specifically, we extract context windows from the training split, downsample each window into three additional resolutions, and feed each scaled input into the pretrained Moirai model to obtain corresponding embeddings $\mathbf{M}$. For each sample at each scale, we form a triplet $(S, X, M)$, where $S$ is the scale index, $X$ is the autocorrelation (ACF) computed on the input, and $M$ is the $\ell_2$ norm of the mean embedding. These triplets enable graph-based causal discovery.

We first apply the PC algorithm [36, 50] with Fisher's $Z$-test ($\alpha = 0.01$). The learned graph includes directed edges $S \rightarrow X$ and $S \rightarrow M$, supporting our assumption that scale causally influences both the input signal and the model representation. To further test whether $S$ acts as a confounder between $X$ and $M$, we compare their raw correlation with the partial correlation conditioned on $S$. The Pearson correlation between $X$ and $M$ is $-0.732$; conditioning on $S$ reduces the partial correlation to $-0.481$ ($p < 0.001$). This reduction indicates that scale partially explains the dependency between $X$ and $M$, consistent with its role as a confounder in our causal formulation. Together, these complementary analyses quantitatively support the causal assumption proposed in Section 3, namely that the scale variable $S$ influences both the observed input $X$ and the model knowledge $M$.

## D    More Experimental Results

### D.1    Further Model Analysis

**Effect of Down-Sampling Methods**    While average pooling is the most commonly used method for generating down-sampled scales, we also investigate two alternative down-sampling techniques to assess their impact. Specifically, we consider *max pooling*, which selects the maximum value within each down-sample kernel, and the *first-step* method, which directly selects the first time step of each kernel. We replace the original average pooling operation in MSFT with these alternatives

and evaluate their performance using MOIRAI_Small on the ETTm1 and ETTm2 datasets. As shown in Figure 6a, the results demonstrate that average pooling consistently outperforms the other two approaches, serving as the most effective method for multi-scale generation.

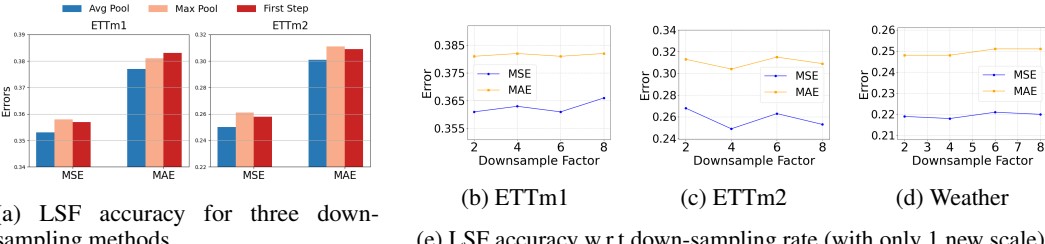

(a) LSF accuracy for three down-sampling methods.

(b) ETTm1     (c) ETTm2     (d) Weather

(e) LSF accuracy w.r.t down-sampling rate (with only 1 new scale).

Figure 6: Overview of LSF accuracy comparison.

**Effect of Down-Sampling Rate**    We investigate the effect of down-sampling rate by using only one new scale ($K = 1$) and comparing the results across different down-sampling factors (2, 4, 6, 8). As shown in Figure 6e. the results reveal that the impact of down-sampling rate varies significantly across datasets. For ETTm1 and Weather, the choice of down-sampling factor is relatively less important, with no single down-sample factor is significantly better than the others. In contrast, ETTm2 exhibits a clear pattern: down-sampling factors of 4 and 8 obviously yield better performance, indicating that the periodic patterns in ETTm2 are better captured with these specific factors. These results demonstrate that the effect of down-sampling is dataset-dependent. Furthermore, they indicate that the performance improvement from using multiple scales is not merely due to adding one particularly important scale but rather results from aggregating information across multiple scales.

**Attention with Aligned Time Indices**    In this section, we provide a detailed discussion of the attention misalignment problem and explore another potential solution. As illustrated in Figure 1 (b) and Figure 5 (a), the problem of directly applying self-attention over the concatenated multi-scale sequence is that cross-scale dependencies are biased to the tokens with the same time ID. However, as tokens in different scales represent various resolution, their time indices do not represent the same temporal location information. The tokens in different scales with the same time id do not correspond to the same temporal range (See 1 (b), left part). Therefore, this time ID-induced bias causes attention to learn misleading temporal correlations.

To address this problem, we test another method based on time id alignment during attention operation. As illustrated in Figure 7, when performing attention between two scales, we map the time ID of tokens in the finer scale to the other coarser scale before RoPE, ensuring that finer-scale tokens from the same temporal range share the same time ID as the corresponding coarse-scale token. Consequently, the resulting attention heatmap in Figure 5 (b) eliminates the cross-scale bias caused by time ID, leading to more reasonable temporal correlations between cross-scale tokens.

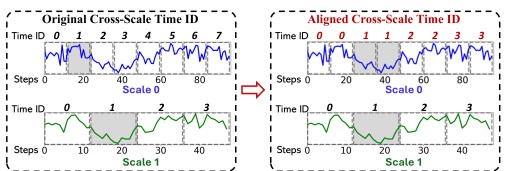

Figure 7: Map the token indices of finer scale to the coarser scale during cross-scale attention

| | ETTm1 | | ETTm2 | | Weather | |
|---|---|---|---|---|---|---|
| | MSE | MAE | MSE | MAE | MSE | MAE |
| Naive | 0.359 | 0.380 | 0.253 | 0.303 | 0.219 | 0.252 |
| Aligned | 0.356 | 0.378 | 0.250 | 0.302 | 0.222 | 0.254 |
| **Ours** | **0.353** | **0.377** | **0.250** | **0.301** | **0.216** | **0.248** |

Table 7: LSF results for MOIRAI_Small using three different attention strategies in MSFT

Table 7 presents the results of the three methods corresponding to the attention patterns in Figure 5. Compared to the Naive method, the time ID alignment approach improves performance on ETTm1 and ETTm2 but shows a performance decline on the Weather dataset. In contrast, our decoupled strategy consistently outperforms both methods. Apart from its inconsistent performance, another limitation of this time ID alignment method is that it is only applicable to Moirai, which employs RoPE for time ID encoding. RoPE allows direct modification of token time IDs during attention, making this adjustment feasible. In contrast, for models using additive position encoding—where

positional information is directly added to each token's input embedding—it is impossible to alter the time ID within the attention blocks. Our method, however, does not rely on modifying time IDs during attention. Instead, it achieves cross-scale alignment through aggregators, making it universally applicable to any model architecture.

**Computation Efficiency**    We compare the memory footprint and training speed of our our methods with the following models: PatchTST [25], iTranformer[19], TimesNet [44], and Scaleformer[34]. For Scaleformer, we follow their original implementation and test it on two backbones Autoformer[45] and Informer[51], referred to as Scaleformer-A and Scaleformer-I, respectively. For MSFT, we test its performance on $\text{MOIRAI}_{\text{Small}}$ and $\text{MOIRAI}_{\text{Base}}$, referred to as MSFT-S and MSFT-B. To ensure a fair comparison, we use a consistent batch size of 32 and a context length of 512 across all models, with a prediction length set to 96. To eliminate external interference, the experiments in this section are exclusively conducted on another server equipped with a 12 vCPU Intel(R) Xeon(R) Platinum 8352V CPU @ 2.10GHz and a single RTX 3080 GPU with 20GB of memory.

The results on ETTm1 and Weather datasets are shown in Table 8. The comparison shows that fine-tuning with MSFT on Moirai does not demand more computational resources than other models. Its GPU memory usage is lower than that of alternative methods. In terms of training speed, MSFT achieves a moderate level among the compared methods. However, it significantly outperform Scaleformer, which is also a multi-scale modeling approach.

Table 8: Quantitative comparison of computation efficiency across different methods.

| Metric | Dataset | PatchTST | TimesNet | iTransformer | Scaleformer-A | Scaleformer-I | MSFT-S | MSFT-B |
|---|---|---|---|---|---|---|---|---|
| Training Speed (ms/iter) | ETTm1 | 65.92 | 334.67 | 27.97 | 307.36 | 180.09 | 103.53 | 185.38 |
| | Weather | 127.67 | 103.20 | 28.90 | 315.44 | 184.05 | 110.80 | 163.93 |
| GPU Memory (MB) | ETTm1 | 4198 | 2786 | 1952 | 11130 | 5104 | 808 | 1916 |
| | Weather | 6866 | 2592 | 2110 | 11138 | 5106 | 808 | 1702 |

We also compare the computational and parameter efficiency of MSFT with several representative fine-tuning strategies, including full finetuning, linear probing, LoRA, and AdaLoRA. Results on $\text{MOIRAI}_{\text{Small}}$with the Weather LSF task are summarized in Table 9.

Table 9: Quantitative comparison of computation efficiency across finetuning methods.

| Method | Params (M) | GPU Mem (MB) | Train Speed (it/s) | Test Speed (it/s) | MSE / MAE |
|---|---|---|---|---|---|
| Full finetuning | 13.8 | 2996 | 9.4 | 104.5 | 0.228 / 0.254 |
| Linear probing | 3.0 | 762 | 19.7 | 113.6 | 0.237 / 0.260 |
| LoRA | 3.4 | 2760 | 11.1 | 115.9 | 0.225 / 0.252 |
| AdaLoRA | 3.4 | 2756 | 10.0 | 102.4 | 0.226 / 0.252 |
| MSFT (ours) | 4.4 | 5616 | 4.6 | 102.5 | **0.216 / 0.248** |

As shown in Table 9, MSFT uses fewer trainable parameters than full finetuning and achieves the best forecasting accuracy (lowest MSE/MAE). The cost is higher GPU memory usage and slower training due to the expanded token length introduced by multi-scale inputs. Nevertheless, its test-time speed remains comparable, offering a reasonable efficiency–performance trade-off compared to other fine-tuning methods.

**More Ablation Study**    To assess robustness, we conducted multiple runs (three seeds) for the experiments in Table 3. Results in Table 11 show that the standard deviations are generally insignificant across settings.

To further validate the role of in-scale masking, we conduct additional ablations where cross-scale aggregators are retained even without masking. Results in Table 10 show that this setting yields worse performance, as attention without in-scale masking learns misaligned cross-scale dependencies, and subsequent aggregation amplifies these inconsistencies. This confirms that cross-scale aggregators must operate jointly with in-scale masking to effectively fuse temporal correlations, while their standalone use significantly compromises performance.

Table 10: Ablation studies on in-scale masking, with mean $\pm$ standard deviation over 3 runs.

| Config | ETTm1 | | ETTm2 | | Weather | |
|---|---|---|---|---|---|---|
| | MSE | MAE | MSE | MAE | MSE | MAE |
| ⑧ | $0.360 \pm 0.003$ | $0.380 \pm 0.002$ | $0.253 \pm 0.000$ | $0.303 \pm 0.000$ | $0.220 \pm 0.000$ | $0.253 \pm 0.001$ |
| ⑧+ C2F | $0.364 \pm 0.004$ | $0.383 \pm 0.003$ | $0.256 \pm 0.002$ | $0.307 \pm 0.002$ | $0.224 \pm 0.000$ | $0.256 \pm 0.000$ |
| ⑧+ F2C | $0.365 \pm 0.003$ | $0.384 \pm 0.002$ | $0.253 \pm 0.000$ | $0.303 \pm 0.001$ | $0.225 \pm 0.001$ | $0.255 \pm 0.001$ |
| ⑧+ Both | $0.365 \pm 0.003$ | $0.385 \pm 0.003$ | $0.255 \pm 0.002$ | $0.305 \pm 0.001$ | $0.226 \pm 0.001$ | $0.256 \pm 0.001$ |
| MSFT | $\mathbf{0.354 \pm 0.003}$ | $\mathbf{0.378 \pm 0.002}$ | $\mathbf{0.250 \pm 0.000}$ | $\mathbf{0.301 \pm 0.000}$ | $\mathbf{0.216 \pm 0.000}$ | $\mathbf{0.248 \pm 0.000}$ |

Table 11: Ablation study on three LSF datasets using MOIRAI$_{\text{Small}}$. Mean $\pm$ standard deviation over 3 runs are reported. MSFT consistently outperforms all ablation variants.

| Config | ETTm1 | | ETTm2 | | Weather | |
|---|---|---|---|---|---|---|
| | MSE | MAE | MSE | MAE | MSE | MAE |
| ① | $0.362 \pm 0.003$ | $0.380 \pm 0.002$ | $0.253 \pm 0.001$ | $0.305 \pm 0.000$ | $0.219 \pm 0.000$ | $0.252 \pm 0.000$ |
| ② | $0.360 \pm 0.002$ | $0.379 \pm 0.002$ | $0.252 \pm 0.000$ | $0.304 \pm 0.000$ | $0.218 \pm 0.000$ | $0.249 \pm 0.000$ |
| ③ | $0.374 \pm 0.004$ | $0.385 \pm 0.003$ | $0.256 \pm 0.001$ | $0.308 \pm 0.001$ | $0.224 \pm 0.002$ | $0.256 \pm 0.001$ |
| ④ | $0.361 \pm 0.002$ | $0.382 \pm 0.002$ | $0.254 \pm 0.000$ | $0.306 \pm 0.000$ | $0.222 \pm 0.001$ | $0.254 \pm 0.001$ |
| ⑤ | $0.371 \pm 0.003$ | $0.384 \pm 0.002$ | $0.256 \pm 0.000$ | $0.307 \pm 0.001$ | $0.223 \pm 0.001$ | $0.255 \pm 0.001$ |
| ⑥ | $0.363 \pm 0.003$ | $0.382 \pm 0.002$ | $0.254 \pm 0.000$ | $0.304 \pm 0.000$ | $0.220 \pm 0.000$ | $0.252 \pm 0.000$ |
| ⑦ | $0.357 \pm 0.002$ | $0.379 \pm 0.001$ | $0.252 \pm 0.000$ | $0.304 \pm 0.000$ | $0.218 \pm 0.000$ | $0.251 \pm 0.000$ |
| ⑧ | $0.360 \pm 0.003$ | $0.380 \pm 0.002$ | $0.253 \pm 0.000$ | $0.303 \pm 0.000$ | $0.220 \pm 0.001$ | $0.253 \pm 0.001$ |
| ⑨ | $0.359 \pm 0.003$ | $0.379 \pm 0.003$ | $0.269 \pm 0.003$ | $0.313 \pm 0.002$ | $0.226 \pm 0.003$ | $0.252 \pm 0.002$ |
| ⑩ | $0.384 \pm 0.007$ | $0.388 \pm 0.004$ | $0.255 \pm 0.003$ | $0.311 \pm 0.002$ | $0.219 \pm 0.001$ | $0.252 \pm 0.000$ |
| MSFT | $\mathbf{0.354 \pm 0.003}$ | $\mathbf{0.378 \pm 0.002}$ | $\mathbf{0.250 \pm 0.000}$ | $\mathbf{0.301 \pm 0.000}$ | $\mathbf{0.216 \pm 0.000}$ | $\mathbf{0.248 \pm 0.000}$ |

## D.2 Evaluation of knowledge forgetting

To further investigate the potential issue of knowledge forgetting, we conduct a simple zero-shot transfer experiment following the setup of [41]. Specifically, we finetune the model on a source dataset $A$ and directly evaluate it on an unseen target dataset $B$, denoted as $A \to B$. Table 12 reports MSE averaged over four prediction lengths for MOIRAI$_{\text{Small}}$. The results reveal no consistent pattern across transfer settings. In some cases (e.g., ETTm1 $\to$ ETTm2), fine-tuning leads to improved zero-shot generalization, while in others (e.g., ETTm2 $\to$ ETTm1) the performance degrades, suggesting that finetuning overrides certain pretrained knowledge. When comparing full finetuning and MSFT, neither method consistently outperforms the other, indicating that both approaches are not explicitly designed to mitigate catastrophic forgetting. Moreover, the multi-scale knowledge learned by MSFT from dataset $A$ may not always generalize to dataset $B$ if their temporal structures differ significantly.

Table 12: Cross-dataset transfer performance (MSE) on MOIRAI$_{\text{Small}}$. Each entry reports the performance when the model is finetuned on dataset $A$ and evaluated on an unseen dataset $B$ (denoted $A \to B$). The results are averaged over four prediction lengths. Best values are highlighted in bold.

| Transfer | Zero-shot | Full FT | MSFT |
|---|---|---|---|
| ETTm1 $\to$ ETTm2 | 0.300 | 0.293 | **0.288** |
| ETTm2 $\to$ ETTm1 | 0.448 | **0.454** | 0.470 |
| ETTm1 $\to$ ETTh1 | 0.416 | **0.410** | 0.420 |
| ETTm2 $\to$ ETTh1 | 0.416 | 0.415 | **0.414** |
| ETTm1 $\to$ ETTh2 | 0.355 | **0.350** | 0.363 |
| ETTm2 $\to$ ETTh2 | 0.355 | 0.359 | **0.350** |

From another perspective, MSFT is inherently more conservative in overwriting pretrained representations, due to its plug-in design. During finetuning, only lightweight adapters, normalization layers, and the output head are updated, while the majority of pretrained weights remain frozen. This design principle is consistent with common strategies in continual learning, where task-specific modules are introduced to reduce forgetting [2]. Furthermore, if users wish to preserve the zero-shot

performance on unseen datasets after finetuning, one can simply deactivate or remove the MSFT modules, effectively reverting the model to its original pretrained TSFM with minimal performance change.

## D.3  Full results

We report the full LSF results on four different prediction lengths, with MSE are shown in Table 13 and MAE are shown in Table 14. Results of deep learning-based baselines are obtained from Liu et al. [19] and Chen et al. [4].

Table 13: Full MSE results of long sequence forecasting experiments.

| Method | ETTm1 | | | | ETTm2 | | | | ETTh1 | | | | ETTh2 | | | | Electricity | | | | Weather | | | |
|---|---|---|---|---|---|---|---|---|---|---|---|---|---|---|---|---|---|---|---|---|---|---|---|---|
| | 96 | 192 | 336 | 720 | 96 | 192 | 336 | 720 | 96 | 192 | 336 | 720 | 96 | 192 | 336 | 720 | 96 | 192 | 336 | 720 | 96 | 192 | 336 | 720 |
| DLinear[2023] | 0.345 | 0.380 | 0.413 | 0.474 | 0.193 | 0.284 | 0.369 | 0.554 | 0.386 | 0.437 | 0.481 | 0.519 | 0.333 | 0.477 | 0.594 | 0.831 | 0.197 | 0.196 | 0.209 | 0.245 | 0.196 | 0.237 | 0.283 | 0.345 |
| PatchTST[2023] | 0.329 | 0.367 | 0.399 | 0.454 | 0.175 | 0.241 | 0.305 | 0.402 | 0.414 | 0.460 | 0.501 | 0.500 | 0.302 | 0.388 | 0.426 | 0.431 | 0.195 | 0.199 | 0.215 | 0.256 | 0.177 | 0.225 | 0.278 | 0.354 |
| iTransformer[2024a] | 0.334 | 0.377 | 0.426 | 0.491 | 0.180 | 0.250 | 0.311 | 0.412 | 0.386 | 0.441 | 0.487 | 0.503 | 0.297 | 0.380 | 0.428 | 0.427 | 0.148 | 0.162 | 0.178 | 0.225 | 0.174 | 0.221 | 0.278 | 0.358 |
| TimeMixer[2024] | 0.320 | 0.361 | 0.390 | 0.454 | 0.175 | 0.237 | 0.298 | 0.391 | 0.375 | 0.429 | 0.484 | 0.498 | 0.289 | 0.372 | 0.386 | 0.412 | 0.153 | 0.166 | 0.185 | 0.225 | 0.163 | 0.208 | 0.251 | 0.339 |
| SimpleTM [2025] | 0.321 | 0.360 | 0.390 | 0.454 | 0.173 | 0.238 | 0.296 | 0.393 | 0.366 | 0.422 | 0.440 | 0.463 | 0.281 | 0.355 | 0.365 | 0.413 | 0.141 | 0.151 | 0.173 | 0.201 | 0.162 | 0.208 | 0.263 | 0.340 |
| MOIRAI_Small | 0.404 | 0.435 | 0.462 | 0.490 | 0.205 | 0.261 | 0.319 | 0.415 | 0.387 | 0.418 | 0.431 | 0.427 | 0.287 | 0.350 | 0.378 | 0.403 | 0.205 | 0.220 | 0.236 | 0.270 | 0.183 | 0.229 | 0.288 | 0.371 |
| + Full finetuning | 0.303 | 0.352 | 0.388 | 0.425 | 0.179 | 0.234 | 0.291 | 0.388 | 0.382 | 0.419 | 0.434 | 0.426 | 0.286 | 0.349 | 0.376 | 0.396 | 0.154 | 0.172 | 0.203 | 0.242 | 0.154 | 0.200 | 0.246 | 0.311 |
| + Linear probing | 0.341 | 0.371 | 0.402 | 0.439 | 0.198 | 0.258 | 0.317 | 0.408 | 0.384 | 0.417 | 0.428 | 0.425 | 0.286 | 0.349 | 0.377 | 0.402 | 0.185 | 0.200 | 0.214 | 0.247 | 0.167 | 0.211 | 0.256 | 0.315 |
| + Prompt tuning | 0.335 | 0.368 | 0.405 | 0.428 | 0.197 | 0.252 | 0.304 | 0.413 | 0.384 | 0.415 | 0.429 | 0.427 | 0.286 | 0.349 | 0.378 | 0.403 | 0.191 | 0.205 | 0.219 | 0.252 | 0.163 | 0.207 | 0.254 | 0.315 |
| + LoRA | 0.302 | 0.357 | 0.389 | 0.431 | 0.179 | 0.234 | 0.288 | 0.387 | 0.382 | 0.418 | 0.431 | 0.426 | 0.286 | 0.349 | 0.377 | 0.402 | 0.152 | 0.176 | 0.197 | 0.243 | 0.153 | 0.197 | 0.243 | 0.305 |
| + AdaLoRA | 0.301 | 0.374 | 0.406 | 0.441 | 0.180 | 0.234 | 0.291 | 0.388 | 0.381 | 0.416 | 0.430 | 0.427 | 0.286 | 0.350 | 0.378 | 0.402 | 0.151 | 0.175 | 0.196 | 0.242 | 0.154 | 0.198 | 0.245 | 0.305 |
| + MSFT | 0.295 | 0.338 | 0.371 | 0.409 | 0.165 | 0.218 | 0.267 | 0.349 | 0.380 | 0.416 | 0.428 | 0.423 | 0.279 | 0.347 | 0.376 | 0.392 | 0.150 | 0.172 | 0.193 | 0.234 | 0.147 | 0.189 | 0.234 | 0.292 |
| MOIRAI_Base | 0.335 | 0.366 | 0.391 | 0.434 | 0.197 | 0.250 | 0.301 | 0.375 | 0.375 | 0.406 | 0.426 | 0.440 | 0.284 | 0.350 | 0.378 | 0.375 | 0.158 | 0.174 | 0.191 | 0.229 | 0.163 | 0.207 | 0.264 | 0.350 |
| + Full finetuning | 0.312 | 0.355 | 0.380 | 0.426 | 0.176 | 0.230 | 0.282 | 0.344 | 0.372 | 0.404 | 0.423 | 0.434 | 0.283 | 0.355 | 0.387 | 0.403 | 0.144 | 0.166 | 0.176 | 0.207 | 0.152 | 0.198 | 0.250 | 0.326 |
| + Linear probing | 0.332 | 0.369 | 0.398 | 0.451 | 0.188 | 0.244 | 0.299 | 0.375 | 0.374 | 0.405 | 0.424 | 0.432 | 0.283 | 0.355 | 0.389 | 0.406 | 0.155 | 0.169 | 0.184 | 0.221 | 0.157 | 0.198 | 0.245 | 0.314 |
| + Prompt tuning | 0.330 | 0.363 | 0.389 | 0.431 | 0.197 | 0.247 | 0.300 | 0.374 | 0.375 | 0.406 | 0.425 | 0.440 | 0.284 | 0.354 | 0.392 | 0.411 | 0.155 | 0.168 | 0.185 | 0.226 | 0.159 | 0.199 | 0.248 | 0.314 |
| + LoRA | 0.311 | 0.345 | 0.373 | 0.414 | 0.177 | 0.230 | 0.280 | 0.347 | 0.373 | 0.404 | 0.423 | 0.434 | 0.284 | 0.351 | 0.379 | 0.411 | 0.142 | 0.163 | 0.178 | 0.210 | 0.151 | 0.198 | 0.249 | 0.322 |
| + AdaLoRA | 0.310 | 0.346 | 0.371 | 0.410 | 0.175 | 0.229 | 0.278 | 0.351 | 0.375 | 0.406 | 0.424 | 0.434 | 0.282 | 0.352 | 0.386 | 0.403 | 0.142 | 0.163 | 0.178 | 0.207 | 0.151 | 0.198 | 0.253 | 0.340 |
| + MSFT | 0.284 | 0.317 | 0.343 | 0.382 | 0.166 | 0.217 | 0.265 | 0.339 | 0.372 | 0.404 | 0.422 | 0.429 | 0.280 | 0.350 | 0.379 | 0.400 | 0.139 | 0.159 | 0.176 | 0.203 | 0.144 | 0.184 | 0.229 | 0.296 |
| MOMENT | - | - | - | - | - | - | - | - | - | - | - | - | - | - | - | - | - | - | - | - | - | - | - | - |
| + Full finetuning | 0.297 | 0.335 | 0.362 | 0.412 | 0.173 | 0.227 | 0.277 | 0.361 | 0.383 | 0.413 | 0.429 | 0.475 | 0.288 | 0.344 | 0.359 | 0.397 | 0.170 | 0.193 | 0.227 | 0.304 | 0.243 | 0.299 | 0.359 | 0.441 |
| + Linear probing | 0.304 | 0.336 | 0.363 | 0.417 | 0.177 | 0.229 | 0.277 | 0.359 | 0.385 | 0.418 | 0.429 | 0.482 | 0.290 | 0.344 | 0.358 | 0.397 | 0.172 | 0.195 | 0.229 | 0.306 | 0.247 | 0.303 | 0.361 | 0.442 |
| + Prompt tuning | 0.302 | 0.339 | 0.366 | 0.415 | 0.176 | 0.229 | 0.279 | 0.359 | 0.386 | 0.416 | 0.429 | 0.478 | 0.289 | 0.345 | 0.361 | 0.398 | 0.172 | 0.194 | 0.228 | 0.304 | 0.244 | 0.299 | 0.360 | 0.441 |
| + LoRA | 0.302 | 0.338 | 0.366 | 0.416 | 0.174 | 0.226 | 0.278 | 0.360 | 0.384 | 0.414 | 0.429 | 0.473 | 0.288 | 0.345 | 0.359 | 0.396 | 0.170 | 0.193 | 0.228 | 0.303 | 0.244 | 0.300 | 0.358 | 0.440 |
| + AdaLoRA | 0.302 | 0.338 | 0.365 | 0.416 | 0.173 | 0.226 | 0.276 | 0.360 | 0.385 | 0.414 | 0.425 | 0.478 | 0.288 | 0.343 | 0.360 | 0.396 | 0.171 | 0.195 | 0.230 | 0.306 | 0.244 | 0.301 | 0.359 | 0.441 |
| + MSFT | 0.289 | 0.327 | 0.354 | 0.404 | 0.170 | 0.222 | 0.273 | 0.356 | 0.381 | 0.410 | 0.426 | 0.469 | 0.286 | 0.341 | 0.358 | 0.394 | 0.166 | 0.190 | 0.226 | 0.300 | 0.237 | 0.297 | 0.356 | 0.438 |
| UNITS | 0.663 | 0.694 | 0.725 | 0.771 | 0.226 | 0.282 | 0.338 | 0.436 | 0.454 | 0.512 | 0.548 | 0.595 | 0.327 | 0.410 | 0.438 | 0.447 | 0.367 | 0.402 | 0.400 | 0.559 | 0.207 | 0.259 | 0.311 | 0.387 |
| + Full finetuning | 0.338 | 0.371 | 0.397 | 0.472 | 0.182 | 0.255 | 0.316 | 0.433 | 0.396 | 0.428 | 0.473 | 0.469 | 0.302 | 0.377 | 0.421 | 0.442 | 0.162 | 0.178 | 0.290 | 0.228 | 0.172 | 0.221 | 0.282 | 0.352 |
| + Linear probing | 0.342 | 0.376 | 0.399 | 0.477 | 0.192 | 0.259 | 0.317 | 0.434 | 0.399 | 0.439 | 0.471 | 0.469 | 0.306 | 0.381 | 0.434 | 0.445 | 0.171 | 0.184 | 0.202 | 0.242 | 0.190 | 0.247 | 0.297 | 0.363 |
| + Prompt tuning | 0.359 | 0.399 | 0.439 | 0.526 | 0.184 | 0.259 | 0.326 | 0.444 | 0.382 | 0.428 | 0.467 | 0.474 | 0.306 | 0.378 | 0.424 | 0.436 | 0.159 | 0.179 | 0.193 | 0.231 | 0.159 | 0.212 | 0.269 | 0.346 |
| + LoRA | 0.338 | 0.370 | 0.396 | 0.466 | 0.183 | 0.256 | 0.315 | 0.431 | 0.377 | 0.426 | 0.463 | 0.481 | 0.300 | 0.379 | 0.422 | 0.433 | 0.163 | 0.170 | 0.192 | 0.228 | 0.163 | 0.214 | 0.274 | 0.349 |
| + MSFT | 0.336 | 0.366 | 0.396 | 0.461 | 0.179 | 0.248 | 0.313 | 0.405 | 0.376 | 0.428 | 0.463 | 0.469 | 0.302 | 0.375 | 0.416 | 0.425 | 0.154 | 0.169 | 0.186 | 0.227 | 0.158 | 0.204 | 0.261 | 0.342 |

Table 14: Full MAE results of long sequence forecasting experiments.

| Method | ETTm1 | | | | ETTm2 | | | | ETTh1 | | | | ETTh2 | | | | Electricity | | | | Weather | | | |
|---|---|---|---|---|---|---|---|---|---|---|---|---|---|---|---|---|---|---|---|---|---|---|---|---|
| | 96 | 192 | 336 | 720 | 96 | 192 | 336 | 720 | 96 | 192 | 336 | 720 | 96 | 192 | 336 | 720 | 96 | 192 | 336 | 720 | 96 | 192 | 336 | 720 |
| DLinear[2023] | 0.372 | 0.389 | 0.413 | 0.453 | 0.292 | 0.362 | 0.427 | 0.522 | 0.400 | 0.432 | 0.459 | 0.516 | 0.387 | 0.476 | 0.541 | 0.657 | 0.282 | 0.285 | 0.301 | 0.333 | 0.255 | 0.296 | 0.335 | 0.381 |
| PatchTST[2023] | 0.367 | 0.385 | 0.410 | 0.439 | 0.259 | 0.302 | 0.343 | 0.400 | 0.419 | 0.445 | 0.466 | 0.488 | 0.348 | 0.400 | 0.433 | 0.446 | 0.285 | 0.289 | 0.305 | 0.337 | 0.218 | 0.260 | 0.297 | 0.348 |
| iTransformer[2024a] | 0.368 | 0.391 | 0.420 | 0.459 | 0.264 | 0.309 | 0.348 | 0.407 | 0.405 | 0.436 | 0.458 | 0.491 | 0.349 | 0.400 | 0.432 | 0.445 | 0.240 | 0.253 | 0.269 | 0.317 | 0.214 | 0.254 | 0.296 | 0.349 |
| TimeMixer[2024] | 0.357 | 0.381 | 0.404 | 0.441 | 0.258 | 0.299 | 0.340 | 0.396 | 0.400 | 0.421 | 0.458 | 0.482 | 0.341 | 0.392 | 0.414 | 0.434 | 0.247 | 0.256 | 0.277 | 0.310 | 0.209 | 0.250 | 0.287 | 0.341 |
| SimpleTM [2025] | 0.361 | 0.380 | 0.404 | 0.438 | 0.257 | 0.299 | 0.338 | 0.395 | 0.392 | 0.421 | 0.438 | 0.462 | 0.338 | 0.387 | 0.401 | 0.436 | 0.235 | 0.247 | 0.267 | 0.293 | 0.207 | 0.248 | 0.290 | 0.341 |
| MOIRAI_Small | 0.383 | 0.402 | 0.416 | 0.419 | 0.282 | 0.318 | 0.355 | 0.410 | 0.398 | 0.423 | 0.435 | 0.450 | 0.334 | 0.374 | 0.395 | 0.421 | 0.299 | 0.310 | 0.323 | 0.347 | 0.216 | 0.258 | 0.297 | 0.346 |
| + Full finetuning | 0.345 | 0.372 | 0.393 | 0.419 | 0.251 | 0.292 | 0.329 | 0.390 | 0.400 | 0.423 | 0.438 | 0.453 | 0.332 | 0.372 | 0.392 | 0.416 | 0.242 | 0.265 | 0.289 | 0.319 | 0.189 | 0.236 | 0.272 | 0.317 |
| + Linear probing | 0.360 | 0.382 | 0.401 | 0.425 | 0.274 | 0.315 | 0.352 | 0.406 | 0.399 | 0.423 | 0.436 | 0.451 | 0.333 | 0.373 | 0.394 | 0.420 | 0.278 | 0.289 | 0.302 | 0.328 | 0.201 | 0.243 | 0.277 | 0.317 |
| + Prompt tuning | 0.359 | 0.380 | 0.403 | 0.423 | 0.273 | 0.309 | 0.343 | 0.409 | 0.402 | 0.423 | 0.435 | 0.451 | 0.337 | 0.373 | 0.394 | 0.420 | 0.285 | 0.294 | 0.307 | 0.331 | 0.199 | 0.241 | 0.276 | 0.317 |
| + LoRA | 0.344 | 0.374 | 0.394 | 0.421 | 0.250 | 0.290 | 0.327 | 0.390 | 0.399 | 0.423 | 0.435 | 0.450 | 0.333 | 0.373 | 0.394 | 0.420 | 0.244 | 0.266 | 0.285 | 0.321 | 0.189 | 0.233 | 0.271 | 0.315 |
| + AdaLoRA | 0.342 | 0.381 | 0.399 | 0.423 | 0.255 | 0.294 | 0.332 | 0.393 | 0.399 | 0.423 | 0.435 | 0.450 | 0.334 | 0.373 | 0.394 | 0.420 | 0.244 | 0.265 | 0.285 | 0.321 | 0.189 | 0.233 | 0.271 | 0.315 |
| + MSFT | 0.341 | 0.367 | 0.387 | 0.414 | 0.242 | 0.281 | 0.314 | 0.368 | 0.401 | 0.421 | 0.433 | 0.449 | 0.326 | 0.369 | 0.391 | 0.413 | 0.241 | 0.262 | 0.282 | 0.316 | 0.185 | 0.229 | 0.266 | 0.311 |
| MOIRAI_Base | 0.360 | 0.379 | 0.394 | 0.419 | 0.271 | 0.306 | 0.339 | 0.388 | 0.398 | 0.417 | 0.429 | 0.452 | 0.334 | 0.380 | 0.405 | 0.432 | 0.248 | 0.263 | 0.278 | 0.307 | 0.198 | 0.240 | 0.282 | 0.338 |
| + Full finetuning | 0.334 | 0.361 | 0.380 | 0.409 | 0.249 | 0.288 | 0.325 | 0.367 | 0.396 | 0.416 | 0.429 | 0.454 | 0.330 | 0.378 | 0.402 | 0.429 | 0.236 | 0.256 | 0.267 | 0.295 | 0.186 | 0.235 | 0.278 | 0.333 |
| + Linear probing | 0.355 | 0.377 | 0.394 | 0.423 | 0.259 | 0.298 | 0.335 | 0.385 | 0.396 | 0.416 | 0.428 | 0.452 | 0.330 | 0.376 | 0.401 | 0.427 | 0.246 | 0.258 | 0.272 | 0.301 | 0.193 | 0.233 | 0.270 | 0.317 |
| + Prompt tuning | 0.355 | 0.377 | 0.393 | 0.417 | 0.271 | 0.301 | 0.339 | 0.387 | 0.397 | 0.416 | 0.428 | 0.452 | 0.335 | 0.378 | 0.404 | 0.432 | 0.246 | 0.258 | 0.274 | 0.305 | 0.196 | 0.235 | 0.272 | 0.318 |
| + LoRA | 0.337 | 0.359 | 0.379 | 0.407 | 0.248 | 0.289 | 0.327 | 0.366 | 0.397 | 0.416 | 0.429 | 0.451 | 0.334 | 0.378 | 0.405 | 0.431 | 0.234 | 0.252 | 0.269 | 0.296 | 0.186 | 0.235 | 0.278 | 0.342 |
| + AdaLoRA | 0.336 | 0.361 | 0.379 | 0.407 | 0.251 | 0.286 | 0.321 | 0.368 | 0.397 | 0.416 | 0.429 | 0.450 | 0.331 | 0.376 | 0.401 | 0.427 | 0.235 | 0.256 | 0.267 | 0.295 | 0.186 | 0.235 | 0.278 | 0.342 |
| + MSFT | 0.335 | 0.359 | 0.378 | 0.404 | 0.246 | 0.285 | 0.320 | 0.369 | 0.395 | 0.415 | 0.429 | 0.450 | 0.327 | 0.374 | 0.404 | 0.427 | 0.230 | 0.252 | 0.266 | 0.293 | 0.182 | 0.224 | 0.261 | 0.311 |
| MOMENT | - | - | - | - | - | - | - | - | - | - | - | - | - | - | - | - | - | - | - | - | - | - | - | - |
| + Full finetuning | 0.348 | 0.369 | 0.386 | 0.415 | 0.262 | 0.299 | 0.332 | 0.386 | 0.406 | 0.424 | 0.445 | 0.485 | 0.348 | 0.386 | 0.404 | 0.437 | 0.276 | 0.292 | 0.313 | 0.363 | 0.240 | 0.287 | 0.328 | 0.384 |
| + Linear probing | 0.353 | 0.370 | 0.385 | 0.414 | 0.265 | 0.300 | 0.333 | 0.385 | 0.406 | 0.429 | 0.445 | 0.490 | 0.350 | 0.387 | 0.404 | 0.437 | 0.278 | 0.294 | 0.314 | 0.364 | 0.246 | 0.290 | 0.330 | 0.386 |
| + Prompt tuning | 0.350 | 0.371 | 0.387 | 0.416 | 0.264 | 0.299 | 0.332 | 0.385 | 0.406 | 0.426 | 0.443 | 0.486 | 0.349 | 0.388 | 0.406 | 0.438 | 0.278 | 0.293 | 0.314 | 0.362 | 0.243 | 0.284 | 0.330 | 0.384 |
| + LoRA | 0.351 | 0.370 | 0.387 | 0.416 | 0.262 | 0.298 | 0.333 | 0.385 | 0.407 | 0.425 | 0.439 | 0.484 | 0.349 | 0.387 | 0.404 | 0.438 | 0.276 | 0.292 | 0.314 | 0.361 | 0.239 | 0.285 | 0.328 | 0.384 |
| + AdaLoRA | 0.351 | 0.370 | 0.387 | 0.415 | 0.262 | 0.298 | 0.331 | 0.386 | 0.407 | 0.426 | 0.438 | 0.487 | 0.348 | 0.386 | 0.405 | 0.437 | 0.277 | 0.292 | 0.314 | 0.364 | 0.242 | 0.288 | 0.329 | 0.385 |
| + MSFT | 0.345 | 0.366 | 0.383 | 0.412 | 0.259 | 0.295 | 0.328 | 0.381 | 0.404 | 0.422 | 0.436 | 0.481 | 0.347 | 0.384 | 0.403 | 0.435 | 0.274 | 0.290 | 0.311 | 0.360 | 0.233 | 0.284 | 0.328 | 0.383 |
| UNITS | 0.520 | 0.541 | 0.561 | 0.588 | 0.301 | 0.333 | 0.367 | 0.420 | 0.444 | 0.478 | 0.495 | 0.547 | 0.362 | 0.412 | 0.441 | 0.455 | 0.438 | 0.467 | 0.465 | 0.582 | 0.254 | 0.294 | 0.328 | 0.376 |
| + Full finetuning | 0.373 | 0.390 | 0.409 | 0.447 | 0.265 | 0.314 | 0.352 | 0.419 | 0.408 | 0.421 | 0.450 | 0.461 | 0.353 | 0.397 | 0.433 | 0.451 | 0.259 | 0.273 | 0.285 | 0.316 | 0.219 | 0.259 | 0.304 | 0.348 |
| + Linear probing | 0.375 | 0.393 | 0.411 | 0.455 | 0.275 | 0.316 | 0.354 | 0.428 | 0.409 | 0.430 | 0.449 | 0.461 | 0.358 | 0.399 | 0.437 | 0.454 | 0.264 | 0.280 | 0.294 | 0.326 | 0.231 | 0.276 | 0.310 | 0.354 |
| + Prompt tuning | 0.391 | 0.411 | 0.436 | 0.482 | 0.270 | 0.318 | 0.359 | 0.431 | 0.399 | 0.427 | 0.447 | 0.460 | 0.347 | 0.395 | 0.430 | 0.446 | 0.259 | 0.277 | 0.290 | 0.321 | 0.206 | 0.256 | 0.295 | 0.347 |
| + LoRA | 0.372 | 0.390 | 0.408 | 0.451 | 0.265 | 0.315 | 0.352 | 0.418 | 0.397 | 0.427 | 0.446 | 0.466 | 0.352 | 0.398 | 0.431 | 0.447 | 0.259 | 0.266 | 0.285 | 0.316 | 0.211 | 0.256 | 0.299 | 0.348 |
| + MSFT | 0.372 | 0.388 | 0.408 | 0.445 | 0.267 | 0.311 | 0.349 | 0.403 | 0.392 | 0.421 | 0.446 | 0.461 | 0.353 | 0.395 | 0.427 | 0.444 | 0.252 | 0.266 | 0.282 | 0.315 | 0.208 | 0.249 | 0.290 | 0.342 |

In addition, for PF, apart from the two metrics we listed in the main text, we demonstrate the results of four additional PF evaluation metrics in Table 15. The baseline results are obtained from Woo et al. [43].

## D.4  Forecast Visualizations

We visualize the forecasting predictions of MSFT using MOIRAI_Small on ETTm1 and ETTm2, with the models finetuned on the predict-96 setup. In addition to the point forecast, which is the median of the samples, the 0.5 and 0.9 quantiles are also plotted for illustration. Only part of the context series is included in the plots.

Table 15: Full results for probabilistic forecasting experiments.

| Method | Electricity | | | | Solar | | | | Weather | | | | Istanbul Traffic | | | | Turkey Power | | | |
|---|---|---|---|---|---|---|---|---|---|---|---|---|---|---|---|---|---|---|---|---|
| | sMAPE | MASE | ND | NRMSE | sMAPE | MASE | ND | NRMSE | sMAPE | MASE | ND | NRMSE | sMAPE | MASE | ND | NRMSE | sMAPE | MASE | ND | NRMSE |
| DeepAR[2020] | 0.118 | 0.844 | 0.080 | 0.704 | 1.385 | 1.222 | 0.520 | 1.033 | 0.776 | 3.170 | 0.163 | 0.486 | 0.249 | 0.613 | 0.139 | 0.181 | 0.404 | 1.395 | 0.083 | 0.181 |
| TFT[2021] | 0.106 | 0.747 | 0.063 | 0.511 | 1.391 | 1.399 | 0.594 | 1.236 | 0.672 | 0.692 | 0.051 | 0.211 | 0.287 | 0.620 | 0.141 | 0.185 | 0.383 | 0.890 | 0.049 | 0.104 |
| PatchTST[2023] | 0.107 | 0.753 | 0.065 | 0.506 | 1.501 | 1.607 | 0.685 | 1.408 | 0.668 | 0.844 | 0.072 | 0.260 | 0.287 | 0.653 | 0.148 | 0.190 | 0.416 | 1.234 | 0.071 | 0.158 |
| TiDE[2023] | 0.102 | 0.706 | 0.061 | 0.514 | 1.400 | 1.265 | 0.538 | 1.093 | 0.636 | 0.832 | 0.066 | 0.214 | 0.280 | 0.618 | 0.140 | 0.185 | 0.389 | 0.904 | 0.059 | 0.139 |
| MOIRAI$_{Small}$ | 0.134 | 0.981 | 0.092 | 0.840 | 1.445 | 1.465 | 0.624 | 1.135 | 0.686 | 0.521 | 0.063 | 0.229 | 0.359 | 0.990 | 0.224 | 0.294 | 0.389 | 0.948 | 0.061 | 0.149 |
| + *Full finetuning* | 0.112 | 0.810 | 0.070 | 1.260 | 1.400 | 1.181 | 0.504 | 1.000 | 0.612 | 0.466 | **0.043** | 0.200 | 0.319 | 0.827 | 0.188 | 0.298 | 0.378 | **0.863** | 0.048 | 0.124 |
| + *Linear probing* | 0.124 | 0.879 | 0.080 | 0.641 | 1.384 | 1.175 | 0.500 | 1.100 | 0.685 | 0.519 | 0.063 | 0.227 | 0.321 | 0.820 | 0.189 | 0.294 | 0.387 | 0.936 | 0.060 | 0.146 |
| + *Prompt tuning* | 0.125 | 0.887 | 0.084 | 0.698 | 1.413 | 1.331 | 0.567 | 1.081 | 0.685 | 0.520 | 0.063 | 0.232 | 0.302 | **0.815** | 0.185 | 0.284 | 0.387 | 0.947 | 0.058 | 0.142 |
| + *LoRA* | 0.123 | 0.872 | 0.079 | 0.650 | 1.391 | 1.160 | 0.495 | 0.953 | 0.617 | 0.472 | **0.043** | **0.197** | 0.327 | 0.907 | 0.206 | 0.282 | 0.382 | 0.887 | 0.055 | 0.131 |
| + *AdaLoRA* | 0.124 | 0.913 | 0.083 | 0.686 | **1.374** | 1.115 | 0.476 | **0.940** | 0.615 | 0.468 | **0.043** | 0.201 | 0.312 | 0.819 | **0.173** | 0.266 | 0.387 | 0.894 | 0.052 | 0.126 |
| + *MSFT* | **0.095** | **0.664** | **0.059** | **0.478** | 1.381 | **1.113** | **0.475** | 0.949 | **0.605** | **0.451** | **0.043** | 0.198 | **0.295** | **0.815** | 0.182 | **0.252** | **0.377** | 0.864 | 0.051 | **0.122** |
| MOIRAI$_{Base}$ | 0.111 | 0.792 | 0.069 | 0.551 | 1.410 | 1.292 | 0.551 | 1.034 | 0.623 | 0.487 | 0.048 | 0.417 | 0.284 | 0.644 | 0.146 | 0.194 | 0.378 | 0.888 | 0.051 | 0.118 |
| + *Full finetuning* | 0.100 | 0.716 | 0.063 | 0.517 | 1.282 | 0.552 | 0.239 | 0.554 | 0.626 | 0.511 | 0.045 | 2.980 | **0.251** | 0.620 | 0.140 | 0.251 | 0.372 | 0.816 | **0.045** | 0.101 |
| + *Linear probing* | 0.109 | 0.776 | 0.070 | 0.603 | 1.387 | 1.212 | 0.516 | 1.021 | 0.620 | 0.480 | 0.048 | **0.203** | 0.302 | 0.574 | 0.130 | 0.180 | **0.256** | 0.949 | 0.053 | 0.120 |
| + *Prompt tuning* | 0.109 | 0.783 | 0.069 | 0.583 | 1.407 | 1.285 | 0.548 | 1.053 | 0.613 | 0.484 | 0.046 | 0.659 | 0.288 | 0.573 | 0.130 | 0.170 | 0.377 | 0.866 | 0.052 | 0.120 |
| + *LoRA* | 0.103 | 0.746 | 0.064 | 0.508 | 1.387 | 1.184 | 0.505 | 0.967 | **0.610** | 0.465 | **0.043** | 0.717 | 0.263 | 0.621 | 0.141 | 0.219 | 0.371 | 0.825 | **0.045** | 0.101 |
| + *AdaLoRA* | 0.108 | 0.779 | 0.068 | 0.561 | 1.405 | 1.186 | 0.506 | 1.010 | 0.613 | **0.456** | 0.044 | 0.417 | 0.281 | 0.660 | 0.149 | 0.194 | 0.376 | 0.875 | 0.047 | 0.102 |
| + *MSFT* | **0.094** | **0.653** | **0.058** | **0.471** | **1.264** | **0.422** | **0.184** | **0.452** | 0.622 | 0.474 | 0.044 | 0.636 | 0.289 | **0.568** | **0.129** | **0.160** | 0.372 | **0.814** | **0.045** | **0.099** |

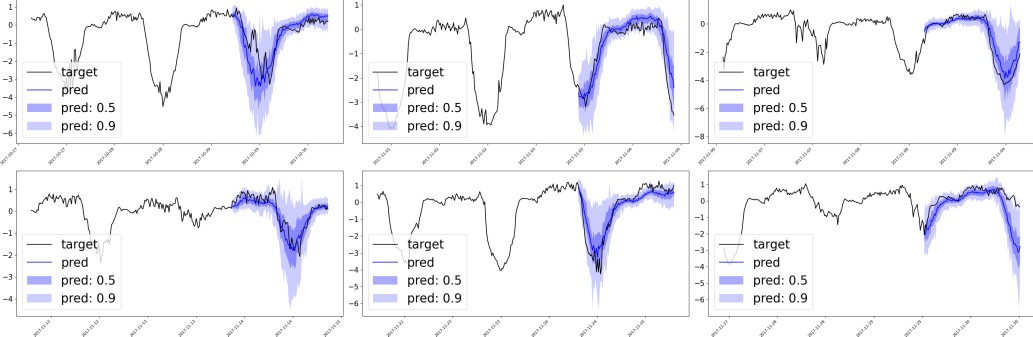

Figure 8: Visualization on ETTm1 (predict-96)

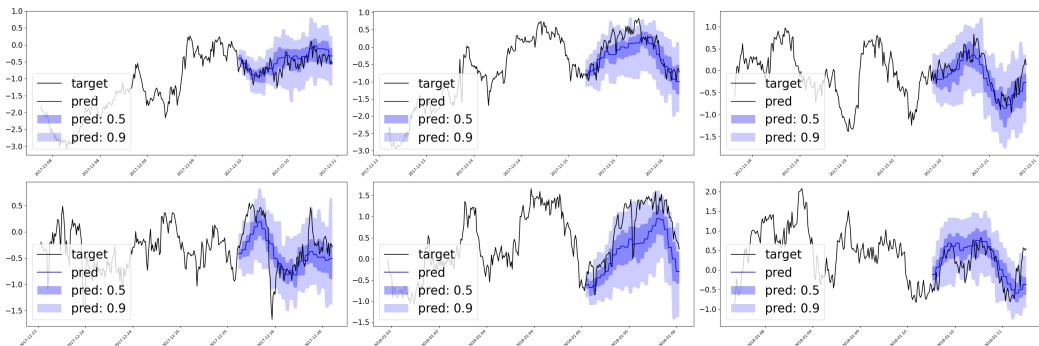

Figure 9: Visualization on ETTm2 (predict-96)

# E    Limitation and Future Work

As indicated in our experiments, MSFT consistently delivers outstanding finetuning results on encoder-based TSFMs, validating the effectiveness of incorporation of multi-scale modeling into TSFM finetuning. However, a natural question that one may be curious about is how to apply our multi-scale finetuning method to TSFMs with other structures, such as decoder-based models.

Here, we first clarify why we focus solely on encoder-based TSFMs in this paper. First, encoder-based models are more flexible to prediction length, making them more efficient to finetune on standard LSF datasets. In contrast, decoder-based models, due to their auto-regressive nature, are significantly slower when finetuning and predicting on long time series. Although some decoder-based models provide finetuning examples, they are often applied to limited datasets without following the standard LSF pipeline. For example, TimesFM are only finetuned on a subset of ETTm dataset for the predict-96 setup. This limitation hinders comprehensive comparisons between our methods and existing LSF baselines or other fine-tuning approaches. Secondly, decoder-based models inherently employ causal masking in their attention mechanisms, which imposes a specific dependency structure. As a

pioneering study, we choose to use encoder-based models without such constraints, providing greater flexibility and generality.

Despite the aforementioned challenges, we provide a potential direction for applying MSFT to decoder-based models. Due to their auto-regressive nature, the causal attention mechanism in decoder-only models can only attend to preceding tokens in the sequence, rather than all tokens simultaneously. Therefore, the creation of multi-scale embedding sequence needs to take the order of scales into account. Similar to Scalerformer[34], we arrange the scales in a coarse-to-fine order and sequentially using coarse information to refine the fine-grained predictions at subsequent levels. First, we concatenate the multi-scale input embeddings as $\boldsymbol{h}_0 = \text{Concat}(\boldsymbol{h}_K^0, \boldsymbol{h}_{K-1}^0, \ldots, \boldsymbol{h}_0^0)$, ensuring the scales are in a coarse-to-fine order. Then, for the attention, we keep using the in-scale masking on the original causal masking, ensuring that the tokens can only attend to the previous tokens from the same scale. Regarding cross-scale aggregators, the original dual-branch design cannot be directly applied due to the auto-regressive nature. Instead, we adopt a single coarse-to-fine branch to fuse the token-level information. The multi-scale mixing remains unchanged, enabling the aggregation of predictions across different scales. We leave the further exploration of this direction as a future work

Another potential limitation is that multi-scale modeling increases the number of input tokens due to the introduction of new scales. Given the transformer's $\mathcal{O}(N^2)$ complexity with respect to input sequence length, this inevitably increases the computational cost. On the other hand, finetuning with new scales can exceed the upper bound of fine-tuning performance achieved on a single scale. Consequently, a trade-off exists between computational cost and performance. Another future direction is to further investigate this trade-off and develop a more efficient strategy to achieve an optimal balance.

