# OpenReview forum: "Multi-Scale Finetuning for Encoder-based Time Series Foundation Models"
_NeurIPS.cc/2025/Conference — NeurIPS 2025 poster_

### Official Review · Reviewer_1tu4 · 2025-06-19

**Clarity:** 4
**Significance:** 3
**Originality:** 3
**Rating:** 4
**Confidence:** 4

**Summary:**

This paper addresses the limitation that current fine-tuning methods for Time Series Foundation Models (TSFMs) often fail to fully exploit the multi-scale characteristics inherent to TSFMs. To this end, the authors propose **MultiScale FineTuning (MSFT)**, a method that captures multi-scale representations of downstream datasets via downsampling. MSFT adopts LoRA to fine-tune the attention computation modules, and explicitly models both intra-scale and cross-scale dependencies. For cross-scale interactions, the method integrates both coarse-to-fine and fine-to-coarse pathways. The final prediction is obtained through a weighted fusion across different scales. The proposed MSFT achieves state-of-the-art fine-tuning performance across multiple TSFMs. Additionally, the authors provide a causal perspective to explain the potential impact of scale on the final prediction outcomes and present corresponding high-level formalizations.

**Questions:**

1. In the causal model proposed in Section 3, the paper lacks empirical validation of the causal graph or model. The current explanation appears more intuitive than analytical. It would be valuable if the authors could design experiments to demonstrate that the scale variable (S) indeed acts as a confounder between the input (X) and the model knowledge (M).
2. In line 190, the paper states: “introduce a set of LoRA modules for each scale.” However, at this stage, the inputs from multiple scales are concatenated. It is unclear how LoRA modules are introduced separately for each scale under this setup. The authors are encouraged to provide more implementation details.
3. It would be helpful if the authors could provide a comparison of MSFT with other fine-tuning methods in terms of parameter count and computational cost across different TSFMs.
4. It is suggested to split Table 3 into three separate tables to present the ablation results more clearly. The current table presents conditions in a complex and unintuitive manner.

**Ethical Concerns:**

["NO or VERY MINOR ethics concerns only"]

**Final Justification:**

Most of my concerns have been addressed.

**Limitations:**

yes

**Quality:**

3

**Strengths And Weaknesses:**

Strengths:

1. The overall writing and logical flow of the paper are well-structured, with clear explanations of the proposed method and experiments.
2. The paper introduces a novel fine-tuning approach for TSFMs that explicitly accounts for the multi-scale nature of time series data.
3. The proposed MSFT method achieves superior fine-tuning performance across multiple TSFMs.
4. The authors provide a theoretical interpretation of MSFT from a causal perspective using causal graphs.

Weaknesses:

1. The current fine-tuning method, MSFT, is designed specifically for encoder-based models, without offering strategies for other types of TSFMs.
2. The paper does not compare the computational or parameter efficiency of MSFT with other fine-tuning methods.

---

> ### Author Rebuttal · Authors · 2025-07-31
>
> We thank the reviewer for the thoughtful and detailed review. We are glad that the reviewer found the writing and logical flow of our paper to be clear and well-structured, and that the proposed method and experiments were easy to follow. We also appreciate the recognition of our novel fine-tuning approach that explicitly leverages the multi-scale nature of time series data. We are particularly encouraged by the reviewer’s acknowledgment of MSFT’s strong empirical performance across multiple TSFMs and the value of our theoretical interpretation from a causal perspective using causal graphs.
>
> We also sincerely thank the reviewer for raising thoughtful questions and suggestions. Below, we provide detailed responses to the weaknesses (W) and questions (Q) raised by the reviewer.
>
> **W1: The current fine-tuning method, MSFT, is designed specifically for encoder-based models, without offering strategies for other types of TSFMs.**
>
> We thank the reviewer for pointing this out. We acknowledge that the current framework design and experimental validation of MSFT are based on encoder-based TSFMs. We choose encoder-based models because they are more efficient to fine-tune than architectures with decoder, making them more suitable for multi-epoch fine-tuning in the popular forecasting benchmark (e.g. LSF datasets). We clearly stated the scope in the title and the paper.  However, the core idea of MSFT is general and can potentially be extended to other types of TSFMs with appropriate adaptations based on their tokenization schemes or attention mechanisms. We have discussed some preliminary strategies in the “Limitation and Future Work” section, and we consider applying MSFT to non-encoder architectures as an important direction for future research.
>
> **W2 & Q3: Request to compare the computational or parameter efficiency of MSFT with other fine-tuning methods.**
>
> Thank you for the valuable comment. We did not include comparisons across other finetuning methods in the manuscripts because our design explicitly introduces multi-scale modeling, which naturally leads to increased input token length and thus extra memory and computational overhead. Therefore, we instead compared with popular deep-learning-based methods and a few baselines that also incorporate multi-scale structure for a more meaningful evaluation.
>
> However, we fully agree with the reviewer that comparing different finetuning strategies on top of TSFMs is also valuable. As such, we conducted additional experiments below, comparing MSFT with full finetuning, linear probing, LoRA, and AdaLoRA, all under the same settings.
>
>
> Method          | Params (M) | GPU Mem (MB) | Train Speed (it/s) | Test Speed (it/s) |  MSE   |  MAE
> ----------------|------------|---------------|---------------------|--------------------|--------|-------
> Full finetuning |    13.8    |     2996      |        9.4         |       104.5        | 0.228  | 0.254
> Linear probing  |     3.0    |      762      |        19.7         |       113.6        | 0.237  | 0.260
> LoRA            |     3.4    |     2760      |         11.1         |       115.9        | 0.225  | 0.252
> AdaLoRA         |     3.4    |     2756      |        10.0         |       102.4        | 0.226  | 0.252
> MSFT (ours)     |     4.4    |     5616      |         4.6         |       102.5        | 0.216  | 0.248
>
> *Table 1: Efficiency analysis of Moirai-Small on Weather LSF task (Prediction length = 96, Batch size = 512)*
>
> As shown, although MSFT uses fewer trainable parameters, it does consume more GPU memory and incurs slower training speed due to the expanded token length from multi-scale inputs. However, test-time speed remains comparable, and importantly, forecasting performance improves (lowest MSE/MAE), highlighting the benefit of our design.
> To summarize, as we have also stated in the Limitation section, multi-scale modeling inherently introduces additional computational overhead due to increased input length and processing complexity. But compared to many deep learning baselines, MSFT still offers a reasonable efficiency–performance trade-off, making it practical and effective for real-world use.
> We hope this clarifies our rationale and we’re happy to add these results into the appendix if helpful.
>
> **Q1: In the causal model proposed in Section 3, the paper lacks empirical validation of the causal graph or model. The current explanation appears more intuitive than analytical. It would be valuable if the authors could design experiments to demonstrate that the scale variable ($S$) indeed acts as a confounder between the input ($X$) and the model knowledge ($M$).**
>
> We thank the reviewer for this insightful comment and valuable suggestion. To empirically validate the proposed causal graph in Section 3, we conduct causal structure learning and partial correlation analysis on ETTm1 dataset across four different scales.
>
> Specifically, we extract context windows from the training split of ETTm1, and downsample each window into three additional resolutions to construct multiscale views. Each scaled input is then passed into the Moirai model to obtain the corresponding input embeddings, which reflect part of the model’s activated knowledge $M$. Since both the scaled input context and the input embeddings vary in length across scales, we construct fixed-length scalar proxy features to enable graph-based causal analysis. For each sample at each scale, we form a (S, X, M) triplet, where S is the scale index (0 to 3), X is the ACF computed on the sample series and M is the L2 norm of the mean input embedding.
>
> We first apply the PC algorithm [1, 2] with Fisher’s Z test (α = 0.01) to the 3-variable dataset, with the goal of identifying whether directed causal edges exist from scale to the input signal and model representation. The inferred structure includes the directed edges scale → acf_sum and scale → emb_l2, supporting the assumption that scale causally influences both the input and the model representation.
>
> To further assess whether scale acts as a confounder between $X$ and $M$, we compare their raw correlation with the partial correlation conditioned on scale. The Pearson correlation between acf_sum and emb_l2 is −0.732. However, after conditioning on scale, the partial correlation reduces to −0.481 (p < 0.001). This reduction in correlation magnitude suggests that scale partially explains the statistical dependency between input and model representation, which is consistent with its role as a confounder in the proposed causal graph.
>
> Together, these two complementary analyses quantitatively support the role of scale as a confounder between input and model representation, consistent with the causal assumptions proposed in Section 3.
>
> [1] An Algorithm for Fast Recovery of Sparse Causal Graphs, 1991
>
> [2] Causal-learn: Causal discovery in python, JMLR 2024
>
> **Q2: How LoRA modules are introduced separately for each scale on the concatenated multi-scale embeddings? The authors are encouraged to provide more implementation details.**
>
> Thank you for the detailed question. Although inputs from multiple scales are concatenated before being fed into the attention layers, we keep track of the token length corresponding to each scale. During the attention computation, we slice the concatenated embeddings to extract the token groups associated with each scale and apply the corresponding LoRA modules accordingly. This allows us to introduce scale-specific LoRA modules even under the concatenated input setup. We also provide a pseudocode in the response to Reviewer oFpM, which the reviewer may refer to for further implementation details.
>
>
>
> **Q4: It is suggested to split Table 3 into three separate tables to present the ablation results more clearly.**
>
> Thank you for the helpful feedback. Due to page limitations, we kept Table 3 in a compact format for the submission version. We will split it into three separate tables in the future version to improve clarity.

---

> > ### Comment · Reviewer_1tu4 · 2025-08-05
> >
> > Thanks for the detailed response and I will keep the positive rate. Good luck!

---

> ### Author Response · Authors · 2025-08-07
> **Thanks for your acknowledgement**
>
> Thank you very much for your acknowledgment and for informing us of your decision!
>
> If you have any remaining concerns or questions, please don’t hesitate to let us know. We would be happy to discuss further and continue improving our manuscript.
>
> Once again, we sincerely appreciate your time and effort in reviewing our paper.
>
> Best regards and best of luck,
>
> The Authors

---

### Official Review · Reviewer_TU3Z · 2025-06-29

**Clarity:** 4
**Significance:** 3
**Originality:** 3
**Rating:** 5
**Confidence:** 5

**Summary:**

This paper proposes MSFT, a novel multi-scale finetuning method for encoder-based Time Series Foundation Models (TSFMs). The authors argue that naive fine-tuning approaches overlook the multi-scale nature of both time series data and TSFMs, leading to suboptimal performance and potential overfitting. Through a causal analysis, they identify scale as a confounding factor between input data and model predictions, and propose MSFT framework, which generates multi-scale inputs via downsampling to block the confounding effect. To address the challenge of scale-varying token resolutions, MSFT employs scale-specific adapters, introduces a decoupled token dependency modeling mechanism, and aggregates predictions across scales using learnable weights. Experimental results on diverse datasets demonstrate that MSFT improves the performance of TSFMs compared with naive fine-tuning and other parameter-efficient fine-tuning methods.

**Questions:**

1. Why is the performance gain of MSFT marginal on the ETTh datasets compared to the ETTm datasets?

2. How is the up-sampling process conducted during multi-scale prediction mixing?

3. Are the models channel-independent?

4. How should the number of scales and downsampling factors be selected to balance performance and computational cost? Would the method benefit by incorporating the prior knowledge of data frequency? For example, would down-sample the minute-level data to hourly data improve efficiency without sacrificing important patterns?

5. Is the proposed method also applicable to TSFMs that do not use patch-based tokenization, such as Chronos [R1] and Time-MOE [R2]?

6. Recent advances in time series foundation models primarily fall into two categories: LLM-based approaches [R3] and models trained from scratch [R4]. This paper adopts the latter strategy—fine-tuning models such as Moirai and UniTS. Could the authors comment on the feasibility and potential advantages or limitations of fine-tuning LLM-based time series models and pre-trained from scratch models? While additional experiments are not necessary, a brief discussion contrasting the two paradigms would be valuable for readers.

[R1]  Chronos: Learning the language of time series. TMLR, 2024.

[R2] Time-moe: Billion-scale time series foundation models with mixture of experts. ICLR 2025

[R3] A survey of time series foundation models: Generalizing time series representation with large language model. arXiv, 2024.

[R4] A survey on time-series pre-trained models. TKDE, 2024.

**Ethical Concerns:**

["NO or VERY MINOR ethics concerns only"]

**Final Justification:**

Thank you for the authors' response, which resolved most of my concerns. I hold a positive view of this work and have decided to keep my overall rating as Accept, while slightly increasing the clarity score.

**Limitations:**

Yes.

**Paper Formatting Concerns:**

None.

**Quality:**

3

**Strengths And Weaknesses:**

Strengths:

1. The experiments are comprehensive. The paper evaluates multiple encoder-based TSFMs on diverse datasets. In addition, it includes various single-scaled baselines for comparison. The experimental results demonstrate the effectiveness of their proposed method.

2. The paper is easy to follow, with a coherent and well-organized  logical structure. Additionally, the research problem of adapting TSFMs to downstream tasks is both timely and essential. Also, multi-scale modeling of TSFMs represents a promising and unexplored direction.

3. The proposed method is simple and easy to understand. Components are well-motivated and aligned with specific challenges. Decoupling within-scale and inter-scale dependencies is novel and reasonable.

Weaknesses:

1. The performance of the proposed method shows less performance improvement on ETTh datasets.

2. Although the authors acknowledge this in the Limitation section, the proposed method requires downsampling the input time series into multiple resolutions and concatenating them, which increases the input sequence length and consequently raises the model's computational cost.

3. Some details in Figure 3 are not clearly explained. For example, the tokens in the cross-scale aggregator are shown in different colors, but the meaning of these colors is not clarified in the text.

---

> ### Author Rebuttal · Authors · 2025-07-31
>
> We sincerely thank the reviewer for the constructive and encouraging feedback. We appreciate the recognition of our comprehensive experimental design, including the evaluation across diverse datasets, multiple encoder-based TSFMs, and strong single-scale baselines. We are also grateful for the acknowledgment of the clarity and organization of our paper, and for identifying the problem of adapting TSFMs to downstream tasks as both timely and important. We are encouraged by the reviewer’s appreciation of the simplicity and motivation behind our method, as well as the novelty of decoupling within-scale and inter-scale dependencies.
>
> We also greatly appreciate the reviewer’s thoughtful questions and constructive suggestions. Below, we provide detailed responses to the weaknesses (W) and questions (Q) raised by the reviewer.
>
> **W1 & Q1: Performance gain of MSFT is marginal on the ETTh datasets compared to the ETTm datasets, why?**
>
> Thank you for the question. We believe there are two main reasons behind this. First, datasets with different original temporal resolutions inherently differ in their capacity to benefit from multiscale modeling. For minute-level data (e.g., ETTm), downsampling by factors like 2/4/8 yields hour-level sequences that still retain clear temporal structures and meaningful patterns. In contrast, for hour-level data (e.g., ETTh), such downsampling offers much less room to generate additional scales with strong periodicity or semantic interpretability.
>
> Second, ETTh datasets are significantly smaller than the ETTm datasets. Since TSFMs typically require long input contexts for effective modeling, the limited data in ETTh makes the models more prone to overfitting, which can reduce the observable benefit from MSFT.
>
> **W2 & Q4: A limitation is the computational overhead from multi-scale input. Is there any guidance on selecting scales and downsampling factor to balance the trade-off?**
>
> Thank you for this practical and insightful question. Incorporating prior knowledge about the raw data frequency into the choice of downsampling factors is indeed a natural and intuitive idea. While most multiscale modeling works (and our main paper) follow the common practice of using powers-of-two downsampling, we also experimented with frequency-informed downsampling strategies, as detailed in “Effect of Down-Sampling Rate”, Appendix C.1. Our results show that frequency-aware downsampling does not consistently bring advantages. It led to noticeable improvement only on ETTm2 (ds_factor = 4, i.e., 15T to 1H), while showing little to no benefit on ETTm1 (ds_factor = 4) and Weather (ds_factor = 6, i.e., 10T to 1H).
>
> We recommend to use 2–3 scales with power-of-two downsampling factors. If GPU memory is limited and a slight drop in performance is acceptable, we suggest using a single scale and treating the downsampling factor as a tunable hyperparameter, including the frequency-informed factor as a candidate value.
>
> **W3: Some details in Figure 3 are not clearly explained, e.g. the meaning of the token's colors is not clarified in the text.**
>
> We thank the reviewer for pointing this out. We will improve Figure 3 and provide additional textual descriptions for the cross-scale aggregator. The color gradient of tokens is intended to illustrate the aggregation across different scales, and we will clarify this in the caption and main text. Additionally, we will include the pseudocode of the aggregation algorithm to help readers better understand the implementation details.
>
>
> **Q2: How is the up-sampling process conducted during multi-scale prediction mixing?**
>
> We experimented with both linear interpolation and repeat-based upsampling, and observed minimal differences in performance on the validation sets. By default, we adopt repeat-based upsampling for its simplicity, but users are free to choose linear interpolation if preferred.
>
> **Q3: Are the models channel-independent?**
>
> The answer depends on the design of the underlying TSFM. Moment is channel-independent, while UniTS adopts variate-wise attention and thus is multivariate. For Moirai, whether to use multivariate modeling is a configurable option. In fact, many datasets in its official implementation adopt the channel-independent setting to achieve the best performance. In our work, we follow the same channel-independent setting for Moirai, including in the zero-shot scenario. This setting is also compatible with our multi-scale implementation, which currently treats different scales as distinct variates. We clarify the channel-independence property of each model in Appendix B.2.
>
> **Q5: Is the proposed method also applicable to TSFMs that do not use patch-based tokenization, such as Chronos and Time-MOE?**
>
> We thank the reviewer for this insightful question. Yes, the proposed MSFT framework can be extended to time series foundation models (TSFMs) that do not rely on patch-based tokenization. Here we focus specifically on the tokenization scheme, and discuss how MSFT can be adapted accordingly.
>
> For example, Time-MoE adopts a point-wise tokenization strategy, which can be treated as using a patch size of 1. Under this view, our multi-scale pipeline can be directly applied without any modification regarding the tokenization process.  For Chronos, the tokenization strategy involves quantizing real-valued time series into discrete tokens by scaling and binning. Each time step is represented by a discrete token, without applying any input projection layers. To adapt MSFT to this setting, we can remove the input projection component used in the original MSFT design and directly apply the same quantization-based tokenization to time series at multiple scales.
>
> Since the scope of this paper focuses on encoder-based TSFMs, and both Chronos and Time-MoE adopt different architectural designs, we did not include experiments on these models. Nonetheless, we discuss potential strategies for extending MSFT to non-encoder architectures in the Limitation and Future Work section. In general, our method is not restricted to patch-based tokenization; it can also be adapted to alternative tokenization schemes.
>
> **Q6: Could the authors comment on the feasibility and potential advantages or limitations of fine-tuning LLM-based time series models and pre-trained from scratch models?**
>
> We thank the reviewer for this valuable open-ended question. Indeed, current advances in time series foundation models (TSFMs) largely fall into two paradigms:
>
> LLM-based TSFMs (e.g., GPT4TS [1], LLM4TS [2], Time-LLM [3]) typically adopt architectures or pretrained checkpoints from large language models. A central focus of these works lies in the embedding strategy, i.e., how to effectively transform time series into embeddings that are compatible with LLM input spaces. One potential advantage of this paradigm is its capacity to naturally incorporate rich contextual or meta-information of the downstream task, which can be helpful in fine-tuning. However, a key challenge is that LLMs are not inherently tailored to capture inductive biases commonly found in time series, such as autocorrelation, periodicity, or hierarchical temporal dependencies. While recent efforts have tried to adapt embeddings and fine-tuning strategies to better suit LLM architectures, it remains difficult to fully preserve the temporal continuity and fine-grained dynamics intrinsic to time series data.
>
> In contrast, pretrained-from-scratch TSFMs (e.g., Moirai, UniTS, TimesFM) are designed and optimized specifically for time series data. These models are pretrained on large-scale univariate or multivariate time series datasets. The main focus is to capture universal temporal patterns or variate-wise correlation by leveraging architectural inductive biases like patching, variate-wise attention, or domain prefix, along with forecasting-aligned pretraining objectives. One clear advantage of this paradigm is that it aligns well with the unique structure of time series data, which helps the model learn fine-grained temporal dynamics more effectively and generalize better across datasets with different sampling rates or sequence lengths. These models also tend to be more lightweight and efficient compared to LLM-based approaches, making them easier to train and deploy in practical settings. On the other hand, they are typically designed for pure time series input, which makes it harder to naturally incorporate external context or metadata, such as text descriptions or user intent.
>
> Considering the respective strengths and limitations of both paradigms, our proposed MSFT framework selects pre-trained-from-scratch models to leverage their general pretrained knowledge on temporal patterns, especially across multiple scales. We would like to investigate applying multi-scale modeling to LLM-based TSFMs as a future direction.
>
> [1] One Fits All: Power General Time Series Analysis by Pretrained LM, NeuriPS 2023
>
> [2] LLM4TS: Aligning Pre-Trained LLMs as Data-Efficient Time-Series Forecast, ACM TIST 2025
>
> [3] Time-LLM: Time Series Forecasting by Reprogramming Large Language Models, ICLR 2024

---

> ### Comment · Reviewer_TU3Z · 2025-08-04
>
> Thank you for the authors' response, which resolved most of my concerns. I hold a positive view of this work and have decided to keep my overall rating as Accept, while slightly increasing the clarity score.

---

> > ### Author Response · Authors · 2025-08-04
> >
> > We thank the reviewer for their time and effort in reviewing our paper. We are encouraged that our work has been positively received by the reviewer, and that most of the raised questions have been clearly addressed. If the reviewer has any remaining questions, please feel free to raise them, and we are happy to discuss further.

---

### Official Review · Reviewer_pUUh · 2025-06-30

**Clarity:** 2
**Significance:** 2
**Originality:** 3
**Rating:** 4
**Confidence:** 4

**Summary:**

The paper identifies a key limitation in the naive fine-tuning of Time Series Foundation Models (TSFMs), specifically their tendency to overfit to a single scale of the data. To address this, the authors propose the MultiScale Fine-Tuning (MSFT) framework, designed for encoder-only TSFMs such as MOIRAI, MOMENT, and UNITS. MSFT enhances performance by fine-tuning on multi-scale inputs. In addition to the core framework, the method incorporates multi-scale mixing, LoRA, and in-scale attention mechanisms into the modified TSFMs. Experimental results demonstrate substantial improvements across benchmarks in both long-sequence forecasting and probabilistic forecasting tasks.

**Questions:**

1. In lines 66–67, the authors claim to be the first to introduce multi-scale modeling in TSFMs. This appears to be inaccurate—see [1] and the Multi-Patch Size Projection layers in MOIRAI.

2. The authors state in lines 155-157: “When directly finetuning the input projection layer over all scales, each scale inherently tends to learn its own specific intra-token patterns, which can lead to interference across scales and suboptimal performance." Wouldn’t this issue also affect the output projection layer? If not, please clarify why.

3. Authors propose in-scale masking to address two challenges:

        a. Lines 153-154: “First, the token schematics and intra-scale dependencies vary significantly across scales.”

        b. Lines 160-164 “Second, standard self-attention introduces misleading cross-scale dependencies due to mismatched time (position) indices.”


both claims are supported only by a single illustrative example. Can the authors provide empirical validation or theoretical justification?

4. Could the authors include extra ablation results in Table 3 showing the impact of removing the in-scale masking component?

5. In Table 3, it would be helpful to visually mark the best and second-best results per column (e.g., boldface, underline) for easier comparison.

**Ethical Concerns:**

["NO or VERY MINOR ethics concerns only"]

**Final Justification:**

The authors have addressed most of my concerns.

**Limitations:**

- The proposed method is limited to encoder-only TSFMs.

- The framework introduces additional computational overhead during both training and inference.

**Quality:**

2

**Strengths And Weaknesses:**

**Strengths:**
- The paper provides a clear identification of the limitations associated with naive finetuning of TSFMs.

- The authors conduct an extensive evaluation, reporting strong empirical performance of their proposed framework.

**Weaknesses:**
- Several claims in the paper lack corresponding empirical support or ablation studies (see questions section for specifics).

- The paper does not report standard deviations in Table 3, making it difficult to assess whether the results are statistically significant or consistent.

- Section 5.2 omits citations to relevant recent work such as [1] and [2], which discuss multi-scale modeling in time series.

- The comparison table is missing key baseline models discussed in Section 5.1, such as [3] and[4].

[1] Vijay Ekambaram, Arindam Jati, Pankaj Dayama, Sumanta Mukherjee, Nam Nguyen, Wesley M. Gifford, Chandra Reddy, Jayant Kalagnanam: “Tiny Time Mixers (TTMs): Fast Pre-trained Models for Enhanced Zero/Few-Shot Forecasting of Multivariate Time Series.” NeurIPS 2024

[2] Shengsheng Lin, Weiwei Lin, Xinyi Hu, Wentai Wu, Ruichao Mo, Haocheng Zhong: “CycleNet: Enhancing Time Series Forecasting through Modeling Periodic Patterns.” NeurIPS 2024

[3] Xiaoming Shi, Shiyu Wang, Yuqi Nie, Dianqi Li, Zhou Ye, Qingsong Wen, Ming Jin: “Time-MoE: Billion-Scale Time Series Foundation Models with Mixture of Experts.” ICLR 2025

[4] Ansari, Abdul Fatir and Stella, Lorenzo and Turkmen, Caner and Zhang, Xiyuan, and Mercado, Pedro and Shen, Huibin and Shchur, Oleksandr and Rangapuram, Syama Syndar and Pineda Arango, Sebastian and Kapoor, Shubham and Zschiegner, Jasper and Maddix, Danielle C. and Mahoney, Michael W. and Torkkola, Kari and Gordon Wilson, Andrew and Bohlke-Schneider, Michael and Wang, Yuyang: “Chronos: Learning the Language of Time Series.” Trans. Mach. Learn. Res. 2024

[5] Shiyu Wang, Haixu Wu, Xiaoming Shi, Tengge Hu, Huakun Luo, Lintao Ma, James Y. Zhang, Jun Zhou: “TimeMixer: Decomposable Multiscale Mixing for Time Series Forecasting.” ICLR 2024

---

> ### Author Rebuttal · Authors · 2025-07-31
>
> We sincerely thank the reviewer for the time and effort dedicated to reviewing our paper. Below, we provide detailed responses to the weaknesses (W) and questions (Q) raised by the reviewer.
>
> **W1-Q1:” The authors claim to be the first to introduce multi-scale modeling in TSFMs. This appears to be inaccurate—see TTM and MOIRAI.”**
>
> We sincerely thank the reviewer for raising this important point. Although multi-scale modeling is a widely used forecasting strategy, we acknowledge that it lacks a universally agreed-upon definition in the time series literature. Furthermore, we recognize that our manuscript lacks a clear and explicit definition of multi-scale modeling, which may have led to ambiguity or misunderstanding when compared with similar concepts such as multi-resolution or multi-patch-size. We appreciate the opportunity to clarify this.
>
> In our context, multi-scale modeling refers to **constructing multiple downsampled versions of a time series instance and jointly leveraging their representations through aggregation during the prediction process**. All works discussed in Section 5.2 follow this paradigm. Below, we clarify the subtle differences between our definition of multi-scale modeling and the related works mentioned by the reviewer:
>
> *TTM* introduces multi-resolution pretraining by downsampling high-frequency datasets into lower-frequency versions, which serves as a form of data augmentation during pretraining. Despite using a learnable resolution prefix and adaptive patching, each time series sample is still processed in a single scale/resolution during its prediction process. Thus, TTM does not perform multi-scale modeling as defined in our work.
>
> *Moirai* uses multiple patch size projection layers for input and output projection. However, it does not downsample the time series to multiple scales/resolutions. Each input series still uses only one patch size during prediction, and all tokens belong to a single temporal scale. While multi-patch-size designs can potentially be used for multi-scale modeling, Moirai itself does not do so. We discuss related works in Appendix A.2.
>
> Therefore, we believe our statement was not overstated. However, we fully understand the reviewer’s concern and are happy to revise the claim to a more precise version: **“the first to introduce multi-scale modeling in TSFM finetuning”**.  We will also revise Section 2 to provide a clearer and more explicit definition of multi-scale modeling.
>
> **W1-Q2: Wouldn’t the multi-scale issue affect the output projection layer? If not, please clarify why.**
>
> We thank the reviewer for this insightful question. In fact, we also experimented with adapting the output head in Moirai into a scale-specific version. However, this led to no performance improvement and significantly increased the parameter count: the output head has a large number of parameters ( ~3M params in Moirai_small has a total of 13.8M), and its mixture distribution nature makes it difficult to apply lightweight adaptations such as adapters.
>
> We posit that this difference arises because the input projection layer lies at the bottom of the model and directly operates on the raw time series, making it more sensitive to the input scale. As it is responsible for learning token-level representations, it plays a crucial role in shaping how subsequent attention and feedforward layers extract dependencies. In contrast, the output head sits at the top of the model and already captures relatively generalizable prediction patterns. As such, further specialization across scales at this stage may have limited effect.
>
> **W1-Q3: Request for empirical validation or theoretical justification regarding following claims**
>
> *(1) "Token schematics vary significantly across scales"*
>
> The intuitive explanation is that tokens across scales correspond to different temporal ranges, leading to differences in their schematics. To quantitatively validate this, we compute the mean cosine similarity of token embeddings between different datasets and scales. Specifically, we sample training sequences from two datasets, ETTm1 (D1) and ETTm2 (D2). Using Moirai’s input projection, we extract embeddings from training sequences of ETTm1 (D1) and ETTm2 (D2) at the original scale (S0) and a downsampled scale (S1, by 2). We compute the mean cosine similarity between the token embeddings from different dataset-scale pairs
>
> | **Pair**              | **Mean Cosine Similarity** |
> |:---------------------:|:--------------------------:|
> | D1 S0 vs. D1 S1 |           0.0974           |
> | D1 S0 vs. D2 S0 |           0.1291           |
> | D2 S0 vs. D2 S1 |           0.1084           |
> | D1 S1 vs. D2 S1 |           0.1447           |
>
> These results show that token embeddings from different scales within the same dataset are less similar than embeddings from the same scale across datasets. This suggests that scale differences even introduce more variation in token schematics than dataset differences, supporting our claim that token semantics vary significantly across scales.
>
> *(2) "Intra-scale dependencies vary significantly across scales"*
>
> The intuitive explanation is that timesteps and tokens across scales correspond to different resolutions, leading to distinct temporal dependencies or dynamics at each scale. To quantitatively support this claim, we compute ACF-based features of the series across different data segments and scales, capturing their temporal dependencies. Specifically, we divide the target series of ETTm1 into two equal-length segments, denoted as D1 and D2. We then compute the ACF and first-order differenced ACF at the original scale (S0) and a downsampled scale (S1). Finally, we calculate the differences in ACF values between various pairs to measure the variation in temporal structures.
>
> | **Pair**             | **Δ ACF** | **Δ Diff1_ACF** |
> |:---------------------|:--------:|:---------------:|
> | D0 S0 vs. D1  S0     | 0.080    | 0.004           |
> | D0 S1 vs. D1  S1     | 0.100    | 0.017           |
> | D0 S0 vs. D0 S1      | 0.390    | 0.044           |
> | D1 S0 vs. D1 S1      | 0.410    | 0.031           |
>
> These results show that scale changes introduce more significant variations in temporal dependencies than changes across time segments, supporting our claim that intra-scale dependencies vary significantly across scales.
>
>
> (3) “Standard self-attention introduces misleading cross-scale dependencies due to mismatched time indices.”
>
> Thank you for the attention to this important point. In fact, we have provided empirical evidence in the manuscript to support this claim. Specifically, both Figure 2 and Section 6.3 visualize attention maps showing that, without in-scale masking, tokens from one scale attend to tokens at the same time index from another scale, even though they represent different actual time range. This results in spurious cross-scale attention that misaligns with true temporal correlations.
>
> We believe these attention maps serve as strong empirical evidence to support our claim. However, if the reviewer finds this insufficient, we would greatly appreciate further suggestions on this.
>
> **W1-Q4: Could the authors include extra ablation results in Table 3 showing the impact of removing the in-scale masking component?**
>
> We thank the reviewer for the suggestion. We would like to clarify that the impact of removing the in-scale masking component is already included in our current manuscript (Row #8 in Table 3). If the reviewer had a different variation or experimental setting in mind, we are happy to further clarify or provide additional results as needed.
>
> **W2: The paper does not report standard deviations in Table 3**
>
> We appreciate the reviewer for pointing this out. As mentioned in our response to Reviewer oFpM (W3 & Q2), we have started running multiple seeds during the rebuttal stage. Preliminary results with standard deviation for several key settings in Table 1 have already been included in the rebuttal, and we will continue expanding them. Due to the large number of experiments and limited computational resources, we were unable to complete all configurations before the rebuttal deadline. We plan to update the results with standard deviations in the follow-up discussion stage, including Table 3 as noted by the reviewer.
>
>
> **W3: Section 5.2 omits citations to relevant recent work (TTM & CycleNet), which discuss multi-scale modeling in time series.**
>
> We thank the reviewer for pointing out these important works. Upon revisiting them carefully, we believe this concern is closely related to the previous question W1-Q1 regarding the definition of multi-scale modeling. We have explained why TTM is not regarded as multi-scale modeling in W1-Q1. For CycleNet, it directly learns a globally shared cycle and forecasts by modeling the residuals after removing them, which does not involve multi-scale modeling.
>
> We acknowledge that both papers represent valuable and timely contributions. We will cite them and add a paragraph in Related Work to explicitly clarify and differentiate these related notions.
>
>
>
> **W4: The comparison table is missing key baseline models (Time-MoE, Chronos)**
>
> We thank the reviewer for raising this point. While the scope of our work is focused on encoder-based TSFMs, we appreciate your valuable feedback. To clarify your suggestion, are you recommending that we (a) include these models as additional baselines in our table (e.g., comparing their zero-shot performance), or (b) apply our MSFT framework to these backbones for a more comprehensive comparison? We would be happy to further investigate this issue in the following discussion stage.
>
> **Q5: Suggestion on Table 3.**
>
> We appreciate the reviewer's suggestion. We have updated Table 3 to highlight the best and second-best results per column using boldface and underlining, respectively.

---

> > ### Author Response · Authors · 2025-08-04
> > **Additional Experimental Results**
> >
> > **W2: The paper does not report standard deviations in Table 3**
> >
> > We have just accomplished multiple runs (3 runs) of experiments presented in Table 3. And we would like to share our results in the following table. To keep the layout clean, we use only the configuration index numbers ( #1– #10 and MSFT) to label each row, consistent with Table 3 of the main paper. For the exact composition of each ablation variant, please kindly refer back to Table 3 in the main submission.
> >
> > | Config | ETTm1        |             | ETTm2        |             | Weather      |             |
> > |--------|--------------|-------------|--------------|-------------|--------------|-------------|
> > |        | MSE          | MAE         | MSE          | MAE         | MSE          | MAE         |
> > | #1     | 0.362 ± 0.003 | 0.380 ± 0.002 | 0.253 ± 0.001 | 0.305 ± 0.000 | 0.219 ± 0.000 | 0.252 ± 0.000 |
> > | #2     | 0.360 ± 0.002 | 0.379 ± 0.002 | 0.252 ± 0.000 | 0.304 ± 0.000 | 0.218 ± 0.000 | 0.249 ± 0.000 |
> > | #3     | 0.374 ± 0.004 | 0.385 ± 0.003 | 0.256 ± 0.001 | 0.308 ± 0.001 | 0.224 ± 0.002 | 0.256 ± 0.001 |
> > | #4     | 0.361 ± 0.002 | 0.382 ± 0.002 | 0.254 ± 0.000 | 0.306 ± 0.000 | 0.222 ± 0.001 | 0.254 ± 0.001 |
> > | #5     | 0.371 ± 0.003 | 0.384 ± 0.002 | 0.256 ± 0.000 | 0.307 ± 0.001 | 0.223 ± 0.001 | 0.255 ± 0.001 |
> > | #6     | 0.363 ± 0.003 | 0.382 ± 0.002 | 0.254 ± 0.000 | 0.304 ± 0.000 | 0.220 ± 0.000 | 0.252 ± 0.000 |
> > | #7     | 0.357 ± 0.002 | 0.379 ± 0.001 | 0.252 ± 0.001 | 0.304 ± 0.000 | 0.218 ± 0.000 | 0.251 ± 0.000 |
> > | #8     | 0.360 ± 0.003 | 0.380 ± 0.002 | 0.253 ± 0.000 | 0.303 ± 0.000 | 0.220 ± 0.001 | 0.253 ± 0.001 |
> > | #9     | 0.359 ± 0.003 | 0.379 ± 0.003 | 0.269 ± 0.003 | 0.313 ± 0.002 | 0.226 ± 0.003 | 0.252 ± 0.002 |
> > | #10    | 0.384 ± 0.007 | 0.388 ± 0.004 | 0.255 ± 0.003 | 0.311 ± 0.002 | 0.219 ± 0.001 | 0.252 ± 0.000 |
> > | MSFT   | **0.354 ± 0.003** |  **0.378 ± 0.002** |  **0.250 ± 0.000**| **0.301 ± 0.000** | **0.216 ± 0.000** | **0.248 ± 0.000** |
> >
> > *Table 1 Ablation study on three LSF datasets using MOIRAI_small*
> >
> > Overall, the variation across multiple runs is not significant, especially for datasets ETTm2 and Weather. Compared to deep-learning-based methods such as TimeMixer (Table 9 in their original paper, std on Weather: 0.01 for MSE, 0.009 for MAE) and TimeMixer++ (Table 13 in their original paper, std on Weather: 0.008 for MSE, 0.007 for MAE), our standard deviations are much smaller. This is because we are fine-tuning a pretrained TSFM, which is less affected by random initialization and typically converges to a relatively stable local minimum during fine-tuning.
> >
> > We can also observe that MSFT consistently outperforms all ablated variants, demonstrating the effectiveness of our method.

---

> > > ### Comment · Reviewer_pUUh · 2025-08-05
> > > **Discussion Response**
> > >
> > > I thank the authors for the detailed response. Their reply addresses most of my concerns, and I will update my rating positively to reflect this. However, I still have a few remaining concerns, outlined below:
> > >
> > > - Regarding Q4: Although row 8 lacks in-scale masking, it also omits C2F and F2C. Given that its performance is very close to MSTF, this raises the question of whether in-scale masking is truly beneficial.
> > >
> > > - For W4: I recommend including results for both zero-shot and naive fine-tuning evaluations of the models.

---

> ### Author Response · Authors · 2025-08-07
>
> We thank the reviewer for their detailed response! We are pleased that most of the concerns have been addressed. For the remaining points, we conducted additional experiments and present the results below.
>
>  > **Q4: Ablation study on in-scale masking**
>
> We appreciate the reviewer’s clarification. In the original ablation, we excluded the cross-scale aggregators when ablating in-scale mask, as we thought the attention mechanism could already capture cross-scale dependencies, making additional aggregators seemingly unnecessary. However, as the reviewer rightly pointed out, a more thorough ablation is helpful for better validating the effect. We thus conducted additional experiments where the aggregators are retained even without in-scale masking.
>
> | Config | ETTm1         |              | ETTm2         |              | Weather       |              |
> |--------|---------------|--------------|---------------|--------------|---------------|--------------|
> |        | **MSE**       | **MAE**      | **MSE**       | **MAE**      | **MSE**       | **MAE**      |
> | #8     | 0.360 ± 0.003 | 0.380 ± 0.002 | 0.253 ± 0.000 | 0.303 ± 0.000 | 0.220 ± 0.000 | 0.253 ± 0.001 |
> | #8 + C2F    | 0.364 ± 0.004 | 0.383 ± 0.003 | 0.256 ± 0.002 | 0.307 ± 0.002 | 0.224 ± 0.000 | 0.256 ± 0.000 |
> | #8 + F2C    | 0.365 ± 0.003 | 0.384 ± 0.002 | 0.253 ± 0.000 | 0.303 ± 0.001 | 0.225 ± 0.001 | 0.255 ± 0.001 |
> | #8 + Both    | 0.365 ± 0.003 | 0.385 ± 0.003 | 0.255 ± 0.002 | 0.305 ± 0.001 | 0.226 ± 0.001 | 0.256 ± 0.001 |
> | **MSFT** | **0.354 ± 0.003** | **0.378 ± 0.002** | **0.250 ± 0.000** | **0.301 ± 0.000** | **0.216 ± 0.000** | **0.248 ± 0.000** |
>
> *Table 2. More ablation studies on in-scale masking*
>
> Notably, adding cross-scale aggregators in this setting leads to even worse results. We posit this is because attention, without in-scale masking, learns misaligned cross-scale dependencies (see Figure 5a). When the aggregators then fuse tokens based on actual temporal positions, they combine inconsistent representations, further amplifying the bias and reducing accuracy.
>
>
> In summary, cross-scale aggregators need to work in conjunction with in-scale masking to correctly fuse temporal correlations. Without in-scale masking, the quality of cross-scale information fusion is significantly compromised.
>
>
> > **W4: Results of Chronos and Time-MoE**
>
> As suggested by the reviewer, we included the results of Chronos (Base) and Time-MoE (Base). The zero-shot results are obtained from Time-MoE paper. For Time-MoE fine-tuning, we fine-tuned the model independently on each dataset. For Chronos, we found that fine-tuning under the LSF setup is extremely time-consuming. Unfortunately, we could not complete it before the discussion deadline. We will continue working on it and include the results in the future version. Below are the current results averaged across four prediction lengths.
>
>
> | Model              | ETTh1        |              | ETTh2        |              | ETTm1        |              | ETTm2        |              | Weather       |              |
> |--------------------|--------------|--------------|--------------|--------------|--------------|--------------|--------------|--------------|---------------|--------------|
> |                    | MSE     | MAE     | MSE    | MAE    | MSE      | MAE   | MSE      | MAE      | MSE     | MAE     |
> | **Chronos (ZS)** | 0.545 |   0.472    | 0.424 | 0.430  | 0.640     |  0.499      | 0.349        | 0.380    |  0.300  |  0.318
> | **Time-MoE (ZS)**  | 0.400        | 0.424        | 0.366        | 0.404        | 0.394        | 0.415        | 0.317        | 0.365        | 0.265         | 0.297        |
> | **Time-MoE (FT)**  | **0.391**        | **0.422**        | 0.353        | 0.395       | 0.369        | 0.389        | 0.297       | 0.348        | 0.236         | 0.284        |
> | **Moirai (MSFT)** | 0.412         | 0.426        | **0.349**        | **0.375**       | **0.354**        | **0.378**        | **0.250**       | **0.301**        | **0.216**       | **0.248**       |
>
> *Table 3. Results of Chronos and Time-MoE, where ZS = zero-shot and FT = full-finetune*
>
> MSFT achieves the best results on **8 out of 10** metrics. Notably, Time-MoE shows very strong performance on ETTh1. We believe this is due to pretraining, which helps capture common patterns in ETTh1. In fact, in their original paper, its zero-shot performance on ETTh1 significantly outperforms the previous SOTA (MSE 0.417), highlighting the benefit of pretraining.
>
> In other words, the performance gap between MSFT and Time-MoE on ETTh1 should not be interpreted as a weakness of our fine-tuning method, but rather reflects the large discrepancy in the zero-shot performance of the pretrained models. As a future direction, we plan to adapt MSFT and apply it to Time-MoE.
>
> In summary, our MSFT still proves to be an effective fine-tuning approach, capable of elevating models with moderate zero-shot ability to a highly competitive performance level.

---

### Official Review · Reviewer_oFpM · 2025-07-01

**Clarity:** 2
**Significance:** 3
**Originality:** 3
**Rating:** 5
**Confidence:** 4

**Summary:**

The paper addresses the problem of fine-tuning a foundation
time series model to a specific dataset, esp. for models that
are based on attention between patches. The authors
propose
-  i) to use adapter layers after the input encoding,
- ii) LoRA modules in the attention layers, and
- iii) let the foundation model process patches at different
    downsampled resolutions, alternating self attention
    only between patches at different times, but the same resolution,
	and between patches at different resolutions, but the same time.

In experiments on three different time series foundation models
and the usual forecasting tasks, the authors show that their
method consistently outperforms other fine-tuning methods
such as full fine-tuning, fine-tuning only the last layer, LoRA etc.

**Questions:**

- q1. Do you rely on a multi-resolution / patch-length property of
  the underlying foundation models? Or do you wrap a multi-resolution
  model around each foundation model, basically feeding it with
  patches in different resolutions?

- q2. Could you also report standard deviations from different runs?
  Which results are significant?

**Ethical Concerns:**

["NO or VERY MINOR ethics concerns only"]

**Final Justification:**

--- added after the rebuttal
- the authors have answered my three issues well.
- it would be good if the material of their answers could be added to the paper, or at
  least its appendix.
- I stay with my anyway positive rating.

**Limitations:**

yes

**Quality:**

2

**Strengths And Weaknesses:**

strong points:
- s1. interesting, timely aspect: fine-tuning time series foundation
  models.
- s2. well engineered, plausible solution: adapter layers, LoRA modules
- s3. novel aspect: feed multi-resolution patches to the time series
  foundation models.
- s4. ample experiments with very good results, improving the state
  of the art.

weak points:
- w1. the formal description of the method is vague.
- w2. the causal modeling motivation is unclear.
- w3. experiments do not report standard deviations.


ad w1. the formal description of the method is vague.
- the method is mostly explained in text and with a big diagram (fig. 3).
  a concise formalization is missing. esp. for the multi-resolution
  aspect I am not sure I understood exactly what the authors
  propose.
- exact pseudocode, e.g., in the appendix, might help a lot.

ad w2. the causal modeling motivation is unclear.
- I do not understand the causal modeling motivation. for example,
  what does "X <- S [...] the input context series X is directly influenced
  by the scale S"?  If X and S denote random variables here, what
  are their domains?

ad w3. experiments do not report standard deviations.
- It is not clear how many runs the authors have conduted.
- As they do not report standard deviations between runs,
  it is not clear which of the differences might be significant.


a few typos and grammar issues:
- line 196 "share the same resolution --- dependencies"
- line 201 "andfine-to-coarse"

---

> ### Author Rebuttal · Authors · 2025-07-31
>
> We sincerely thank the reviewer for the thoughtful feedback and positive evaluation of our work. We are glad that the reviewer finds the topic of fine-tuning TSFMs both timely and practically significant. We also appreciate the recognition of our paper as well-completed, with comprehensive experiments and clear, well-structured writing. We are especially encouraged that the reviewer considers our method to be novel, well-designed, and reasonable, and acknowledges the strength of our empirical results.
>
> We are especially thankful for the valuable comments and questions raised. Below, we provide detailed responses to the weaknesses (W) and questions (Q) raised by the reviewer.
>
> **W1: The formal description of the method is vague. Adding pseudocode would improve clarity.**
>
> We appreciate the valuable suggestions from the reviewers. We agree adding such a pseudocode can greatly improve the clarity. While we experimented with LaTeX-style pseudocode in the OpenReview system, we found it cumbersome and quite difficult to read. Therefore, as a temporary solution, we decided to show our pseudocodes in a Python-like version to convey the core ideas more clearly.
>
> * Overall Training Pipeline for MSFT:
>
> ```
> """
> Input / Hyper-parameters:
>     X, Y                    # input context and prediction windows
>     K                       # number of scales
>     s = 2                   # downsample factor
>     P                       # patch size
>     L                       # number of MSFT layers
> """
>
> # Step 1: Multi-Scale Generation
> for i = 0 to K:
>     X_i = AvgPool(X, window_size = s^i)       # pre-pad X if len(X) % P ≠ 0
>     Y_i = AvgPool(Y, window_size = s^i)       # post-pad Y if len(Y) % P ≠ 0
>     S_i = (X_i, Y_i)
>
> # Step 2: Patching and Scale-specific Input Projection
> H_0 = []
> for i = 0 to K:
>     x_i = Patching(S_i, patch_size = P)
>     h_i = Linear_i(InProject(x_i))            # frozen InProject
>     h_i = Masking(h_i)                        # mask prediction tokens
>     H_0.append(h_i)
>
> h_0 = Concat(H_0)                              # concatenate all scales
> scale_index = GetScaleIndex(H_0)              # index range for each scale in h_0
>
> # Step 3: Multi-Scale Attention Encoding
> h_l = h_0
> for l = 1 to L:
>     h_l = AttnBlock_with_MSFT(h_l, scale_index)
>
> # Step 4: Output Projection and Loss Computation
> H_L = Split(h_l, scale_index)                 # recover [h_0^L, ..., h_K^L]
> losses = []
> for i = 0 to K:
>     Y_hat_i = OutProject(H_L[i])
>     L_i = Loss(Y_i, Y_hat_i)
>     losses.append(w_i * L_i)                  # weighted by learnable w_i
>
> L_total = sum(losses)
> ```
>
> * Multi-Scale Attention Block (AttnBlock_with_MSFT):
>
> ```
> """
> Input:
>     h_in                                  # concatenated multi-scale token embeddings
>     scale_index = [idx_0, ..., idx_K]     # index ranges for each scale
>     s                                     # downsampling factor (e.g., s = 2)
>
> Initialization / Parameters:
>     F2CMap = [Linear_0→1, ..., Linear_{K-1}→K]   # fine-to-coarse projections
>     C2FMap = [Linear_1→0, ..., Linear_K→{K-1}]   # coarse-to-fine projections
> """
>
> # Step 1: Split input into scale-wise representations
> H_in = [h_in[..., idx, :] for idx in scale_index]     # H_in = [h_0, h_1, ..., h_K]
>
> # Step 2: Scale-specific Attention with LoRA
> Q, K, V = [], [], []
> for i = 0 to K:
>     Q_i = W_Q_i(H_in[i])
>     K_i = W_K_i(H_in[i])
>     V_i = W_V_i(H_in[i])
>     Q.append(Q_i)
>     K.append(K_i)
>     V.append(V_i)
>
> # Step 3: In-scale masked attention
> H_attn = ScaledDotProductAttention(Q, K, V, attn_mask = M_in)
>
> # Step 4: Cross-scale Aggregation
>
> # (a) Coarse-to-Fine (C2F); Eq 3
> H_c2f = H_attn.copy()
> for i = K to 1:
>     h_proj = C2FMap[i-1](H_attn[i])
>     H_c2f[i-1] = H_c2f[i-1] + Repeat(h_proj, repeat_factor = s)
>
> # (b) Fine-to-Coarse (F2C);Eq 4
> H_f2c = H_attn.copy()
> for i = 0 to K - 1:
>     h_proj = F2CMap[i](H_attn[i])
>     H_f2c[i+1] = H_f2c[i+1] + AvgPool(h_proj, pool_size = s)
>
> # (c) Merge embeddings from two branches
> H_out = []
> for i = 0 to K:
>     H_out.append((H_f2c[i] + H_c2f[i]) / 2)
>
> # Step 5: Re-concatenate and project
> h_out = Concat(H_out)
> h_out = W_out(h_out)
>
> # (Add & Norm → FeedForward → Add & Norm omitted for brevity)
>
> Return: h_out
> ```
>
> We will include a more formal, academic-style algorithm description in the revised manuscript.
>
> **W2:  The causal modeling motivation is unclear. Why Input context series X is directly influenced by the scale S? Are they random variables? If yes, what are their domains?**
>
> We thank the reviewer for the insightful question. We take this opportunity to clarify our causal modeling motivation and discuss our understanding of time series from a data generation perspective.
>
> Fundamentally, a time series is a discretized sequence of sampled observations derived from an underlying continuous process. Under this view, the input context window $X$ is effectively a discrete observation (in the form of time series) from the true process that occurred during the context period. The variable $X$ arises from two factors: the latent continuous process $I$ (which we do not observe), and the scale parameter $S$, which governs the sampling resolution. $S$ determines how densely the underlying process is sampled, thus shaping both the temporal granularity and the length of the observed sequence $X$. While our manuscript omits the latent variable $I$ for tractability, the edge $S \rightarrow X$ in our causal graph is intended to capture this observation mechanism. Note that scale influences the form of the observed input time series, though not the underlying process itself.
>
> Building on the above discussion, we answer the following questions. Yes, both $S$ and $X$ are treated as random variables. We define the scale variable $S$ as a discrete random variable that selects a resolution level to sample the input (in practice, it is downsampling the original given context time series). $S$ takes values in a finite index set, i.e., $S \in \mathcal{S} =$ { $s_0, s_1, \dots, s_K $ }, where each scale level $s_k$ corresponds to a specific downsampling factor or temporal resolution.
>
> We define $X$ as a random variable that represents the observed input time series segment corresponding to the context period, whose length depends on the scale variable $S$. Formally, we have $X \in \mathcal{X}$, where $\mathcal{X} = \bigcup_{s \in \mathcal{S}} \mathbb{R}^{L_s}$. Here, $L_s$ is the input sequence length corresponding to scale $s \in \mathcal{S}$.
>
> We also analyze the causal model in the response of Q1, Reviewer 1tu4. If the reviewer has remaining concerns, we would be happy to further discuss.
>
> **W3 & Q2: Experiments do not report standard deviations.**
>
> We appreciate the reviewer for raising this point. During the rebuttal period, we have started running multiple seeds for key configurations. As an initial step, we report below the multi-run results of Full FT and MSFT on Moirai-small. For each experiment, we run 3 times with different random seeds and report the average MAE and MSE over four prediction lengths. Most results align with our original results and the standard deviations are generally insignificant across settings.
>
> | Method   | ETTm1          | ETTm2          | ETTh1          | ETTh2          | Electricity     | Weather         |
> |----------|----------------|----------------|----------------|----------------|------------------|------------------|
> | Full FT  | 0.369 ± 0.004  | 0.274 ± 0.004 | 0.416 ± 0.002  | 0.352 ± 0.002  | 0.194 ± 0.004    | 0.229 ± 0.005    |
> | MSFT     | 0.354 ± 0.003  | 0.250 ± 0.000  | 0.412 ± 0.001  | 0.349 ± 0.001  | 0.187 ± 0.001    | 0.216 ± 0.000    |
>
> *Table 1. Multi-run MSE results*
>
> | Method   | ETTm1          | ETTm2          | ETTh1          | ETTh2          | Electricity     | Weather         |
> |----------|----------------|----------------|----------------|----------------|------------------|------------------|
> | Full FT  | 0.383 ± 0.003  | 0.317 ± 0.002  | 0.429 ± 0.001  | 0.379 ± 0.002  | 0.279 ± 0.001    | 0.255 ± 0.003   |
> | MSFT     | 0.378 ± 0.002  | 0.301 ± 0.000  | 0.426 ± 0.001  | 0.375 ± 0.001  | 0.275 ± 0.001    | 0.248 ± 0.000    |
>
> *Table 2. Multi-run MAE results*
>
> As we acknowledged in the checklist, we did not perform multiple runs in the main submission due to computational constraints, similar to Time-MoE and Moment. For example, our LSF evaluation involves 4 backbones × 5 fine-tuning methods × 6 datasets × 4 prediction lengths, totaling 480 experiments. Given this scale, running multiple seeds was not feasible during the rebuttal period. We plan to continue running additional runs for other important settings and will include the full results with standard deviations in a follow-up discussion and future version of the paper.
>
> **Q1: Do you rely on a multi-resolution / patch-length property of the TSFM? Or do you wrap a multi-resolution model around each TSFM?**
>
> We do **not** rely on a multi-resolution or multi-patch-length property of the underlying TSFM. This is because not all TSFMs inherently possess such architectural designs. For instance, while Moirai adopts multiple patch sizes, other models like Moment and UniTS do not. Our goal is to build a general framework that can be applied across different encoder-based TSFMs, regardless of their specific architectures.
>
> Therefore, your second interpretation is more accurate:  our MSFT Framework effectively serves as a wrapper that transforms any TSFM into a multi-scale version by feeding it inputs at multiple temporal resolutions. Specifically, the patch size is fixed, but patches from different scales correspond to different temporal resolutions (see Figure 2b). In summary, what we rely on is not specific architectural properties; instead, we reply on the general pretrained knowledge that the TSFM has acquired, which we hypothesize contains rich multiscale temporal information.
>
> **Typos and grammar issues**
>
> Thank you very much for pointing these out. We have corrected the typos in the manuscripts.

---

> ### Author Response · Authors · 2025-08-07
> **Follow-up on Rebuttal and Clarifications**
>
> Dear Reviewer oFpM
>
> Thank you again for your time and encouraging feedback. As the discussion phase is approaching its end, we would like to kindly follow up and hear your thoughts regarding our rebuttal. We hope that our response has adequately addressed your concerns, and we are more than willing to provide further clarification if needed.
>
> For your convenience, we briefly summarize our responses below:
>
> * W1: We included pseudo codes of our method to further improve the clarity.
>
> * W2: We explained why input context series X is directly influenced by the scale S in our causal model, and clarified their domains as random variables.
>
> * W3: We reported the standard deviations of key configurations, demonstrating that randomness plays a relatively minor role in the fine-tuning performance.
>
> * Q4: Our method can be viewed as wrapping a multi-resolution structure around each TSFM, without relying on any specific property of its architecture.
>
> We sincerely thank you for carefully reading our paper and raising these insightful questions. Your feedback is very valuable to us. Please let us know if any points remain unclear. We would be happy to continue the discussion and further improve the paper.
>
> Best regards,
>
> The Authors

---

### Official Review · Reviewer_7Ko1 · 2025-07-05

**Clarity:** 3
**Significance:** 3
**Originality:** 2
**Rating:** 3
**Confidence:** 3

**Summary:**

This paper focuses on the task of fine-tuning time series foundation model. To address the issues of overfitting and suboptimal performance of naive fine-tuning methods, the paper develops a diversified fine-tuning approach based on different sampling scales. Through extensive experiments, the paper emphasizes the importance of explicitly modeling multiple scales and highlight the shortcomings of simpler methods.

**Questions:**

	It is recommended to provide an experiment to demonstrate how the new method better addresses the overfitting issue, such as by showing the trend of the loss curves.
	It is suggested to add a set of zero-shot experiments to compare the predictive ability of the model before and after fine-tuning on unseen datasets.

**Ethical Concerns:**

["NO or VERY MINOR ethics concerns only"]

**Limitations:**

Some

**Quality:**

3

**Strengths And Weaknesses:**

Strengths:
	The paper addresses a highly relevant topic on effectively fine-tuning time series foundation models, aligning well with practical needs and significance.
	The paper is well-completed, using comprehensive main experiments, analysis, and ablation studies to clearly demonstrate the rationale and utility of each component.
	The writing is logically structured and clearly articulated.
	The paper takes a novel approach by introducing in-scale and cross-scale plugins to address issues like time-index confusion in multi-scale joint modeling, which is an interesting concept.
Weakness:
	The paper mentions that naive fine-tuning strategies can lead to overfitting; however, it does not provide comparative experiments to support this claim.
	Another critical aspect of fine-tuning tasks is preventing knowledge forgetting, and this work appears to lack analysis on this issue.

---

> ### Author Rebuttal · Authors · 2025-07-31
>
> We thank the reviewer for the constructive feedback and encouraging comments. We are glad that the reviewer finds the topic of fine-tuning time series foundation models (TSFMs) to be timely and practically significant. We also appreciate the recognition of our efforts in presenting a well-completed manuscript with comprehensive experiments, as well as the acknowledgment of the clarity in our writing and the logical structure of the paper. We are especially encouraged that the reviewer considers our proposed method to be novel, well-designed, and reasonable.
>
> We also sincerely appreciate the valuable questions and suggestions raised. We now provide detailed responses to the two specific weaknesses (W) and questions (Q) raised by the reviewer.
>
> **W1&Q1: Request for empirical validation of overfitting and how MSFT mitigates it**
>
> We thank the reviewer for the valuable suggestion.  To further illustrate the overfitting phenomenon, we use Moirai-Small to compare Full Finetuning, Linear Probing (LP), LoRA, and MSFT on two datasets: ETTm1 and Weather. Since figures are not allowed in the rebuttal, we present key training NLL loss and validation MSE loss values in a table format at three critical checkpoints:
> * **Epoch1**: after training for one epoch
> * **Epoch***: the epoch when early stopping is triggered
> * **Last**: three epochs after epoch* (i.e., the patience of early stopping)
>
> In our context, overfitting is reflected not only by the upward trend of validation loss after a certain point, but also by the model's tendency to overfit to a single scale, resulting in suboptimal validation performance. Therefore, we analyze both the loss trend after Epoch* and the validation loss value at Epoch*. The results are summarized below:
>
>
> |                | ETTm1 (Epoch1) | ETTm1 (Epoch*) | ETTm1 (Last) | │ | WTH (Epoch1) | WTH (Epoch*) | WTH (Last) |
> |:--------------:|:------------:|:------------:|:----------:|:-:|:-------------:|:------------:|:----------:|
> | Full           |   0.557      |    0.438     |   0.412    | │ |   -0.644      |   -0.758      |  -1.080    |
> | LP             |   0.524      |    0.497     |   0.471    | │ |   -0.366      |   -0.445      |  -0.446    |
> | Lora           |   0.630      |    0.448     |   0.438    | │ |   -0.373      |   -0.851      |  -0.873    |
> | MSFT           |   0.614      |    0.467     |   0.458    | │ |   -0.144      |   -0.615      |  -0.633    |
>
> *Table 1. Train NLL loss on ETTm1 and Weather (WTH) at different checkpoints*
>
>
>
> |  | ETTm1 (Epoch1) | ETTm1 (Epoch*) | ETTm1 (Last) | │ | WTH (Epoch1) | WTH (Epoch*) | WTH (Last) |
> |:------------:|:------------:|:------------:|:----------:|:-:|:-----------:|:----------:|:--------:|
> | Full         |   0.4000     |    0.385     |   0.391    | │ |   0.398     |   0.394     |  0.408   |
> | LP           |   0.444      |    0.408     |   0.408    | │ |   0.421     |   0.402     |  0.403   |
> | Lora         |   0.431      |    0.391     |   0.393    | │ |   0.414     |   0.394     |  0.396   |
> | MSFT         |   0.428      |    0.384     |   0.384    | │ |   0.398     |   0.366     |  0.366   |
>
> *Table 2. Validation MSE loss on ETTm1 and Weather (WTH) at different checkpoints*
>
> We observe the following patterns:
>
> * **Training loss**: All methods show a consistent decrease in training loss throughout training. Note that the training losses are negative log-likelihood (NLL) values based on predictive distributions, and can therefore be negative. Also, since MSFT uses a weighted sum of losses on multiple scales, its training loss is not directly comparable to single-scale baselines.
>
> * **Validation loss**: We monitor the validation MSE loss for clearer comparison across methods, as it aligns with the evaluation metric used on the test set.
>
>   - Among the single-scale baselines, Full Finetuning achieves the lowest validation loss at Epoch*, but the loss quickly increases afterward, forming a clear V-shaped curve, indicating that full finetune is prone to overfitting.
>
>   - Compared to Full Finetuning, Linear Probing and LoRA exhibit more stable loss after Epoch*, with only mild increases. However, due to limited trainable parameters, LP shows noticeably higher loss than both Full Finetuning and LoRA. The performance of LoRA is generally comparable to Full Finetuning in terms of loss magnitude.
>
>   - In contrast, MSFT not only maintains stable loss after Epoch*, but also achieves lower validation loss than the single-scale Full Finetuning, indicating that MSFT mitigates overfitting to a single scale and achieves better performance by using potential temporal resolutions.
>
> We will **include the full loss curves in the revised manuscript** for better visualization.
>
> **W2&Q2: Request for evaluation of knowledge forgetting via zero-shot transfer experiments**
>
> We thank the reviewer for raising this interesting question. We agree that our current manuscript does not include experiments specifically targeting knowledge forgetting.
>
> First, we would like to clarify that the primary focus of this work is on fine-tuning for a single downstream task, aiming to fully activate the pretrained knowledge of a TSFM on a specific dataset.  Given this focus, the continual learning paradigm was not considered necessary in our main experimental design.
>
> However, we agree that preventing knowledge forgetting is an intriguing direction. To explore the adaptation ability of MSFT, we follow the setup of TimeMixer++ [1] and conduct a simple cross-dataset evaluation using Moirai-small. We finetune the model on dataset A and evaluate it on the unseen dataset B, denoted as A → B. The MSE results averaged over four prediction lengths are summarized below:
>
>
>
> |    | Zero-shot | Full FT | MSFT |
> |:------------|:---------:|:-------:|:----:|
> | ETTm1 → ETTm2     | 0.300     | 0.293   | **0.288** |
> | ETTm2 → ETTm1     | **0.448** | 0.454   | 0.470     |
> | ETTm1 → ETTh1     | 0.416     | **0.410** | 0.420   |
> | ETTm2 → ETTh1     | 0.416     | 0.415   | **0.414** |
> | ETTm1 → ETTh2     | 0.355     | **0.350** | 0.363   |
> | ETTm2 → ETTh2     | 0.355     | 0.359   | **0.350** |
>
> *Table 3. Transfer performance (MSE) across datasets. Best values are in bold.*
>
> The results in the table indicate that there is no consistent pattern across all transfer settings. In some cases, knowledge forgetting occurs. The performance of the model on the unseen dataset B after fine-tuning on dataset A is worse than its original zero-shot performance, suggesting that fine-tuning overrides the pretrained knowledge. In other cases, we observe positive knowledge transfer. The model fine-tuned on dataset A achieves better zero-shot performance on dataset B compared to the original pretrained model, indicating that certain knowledge acquired from dataset A generalizes and improves forecasting on dataset B. When comparing Full FT and MSFT, there is no clear winner overall. Both methods do not explicitly address catastrophic forgetting. Moreover, the multi-scale knowledge learned by MSFT from dataset A may not generalize well to dataset B, especially if the temporal structures differ significantly.
>
> From another perspective, **MSFT is fundamentally effective in preserving pretrained knowledge, due to its plug-in design**. MSFT introduces lightweight adapters to learn new tasks, while **keeping the vast majority of pretrained weights frozen** during fine-tuning. Among the pretrained components, only the normalization layers and the output projection head are updated. This design aligns with a common strategy in continual learning that adopts task-specific architectural modules to avoid catastrophic forgetting [2].
>
> Therefore, if users wish to retain the zero-shot performance or the pretrained knowledge on unseen datasets after fine-tuning, they can simply **deactivate or remove all MSFT-related modules**, including multi-scale modeling and aggregation mechanisms. In this case, the model reverts to its original single-scale version, with performance closely resembling that of the pretrained TSFM without fine-tuning.
>
> [1] TimeMixer++: A General Time Series Pattern Machine for Universal Predictive Analysis, ICLR 2025
>
> [2] A Comprehensive Survey of Continual Learning: Theory, Method and Application, TPAMI 2024

---

> ### Author Response · Authors · 2025-08-07
> **Follow-up on Rebuttal and Clarifications**
>
> Dear Reviewer 7Ko1,
>
> Thank you again for your time and valuable comments. As the discussion phase is approaching its end, we would like to kindly follow up and hear your thoughts regarding our rebuttal. We hope that our response has adequately addressed your concerns, and we are more than willing to provide further clarification if needed.
>
> For your convenience, we briefly summarize our responses below:
>
> * W1 & Q1: We demonstrated the overfitting issue of naïve fine-tuning by comparing training and validation losses. We will include the corresponding loss curves in the revised manuscript.
>
> * W2 & Q2: We conducted the transfer learning experiment exactly as you suggested — that is, by "comparing the predictive ability of the model before and after fine-tuning on unseen datasets".  We also clarified that MSFT can effectively mitigate forgetting by removing the introduced fine-tuning modules.
>
> We sincerely thank you for carefully reading our paper and raising these insightful questions. Your feedback and suggestions are important for us. Please let us know if any points remain unclear. We would be happy to continue the discussion and further improve the paper.
>
> Best regards,
>
> The Authors

---

### Author Response · Authors · 2025-08-08
**Summary of Reviewer Feedback and Our Responses**

Dear Reviewers and AC,

As the discussion stage is approaching its end, we would like to take this opportunity to summarize the key reviewer feedback and our responses.  First, we would like to thank all five reviewers for their time and constructive feedback. **Reviewer 7Ko1 (R1)**, **Reviewer oFpM (R2)**, **Reviewer pUUh (R3)**, **Reviewer TU3Z (R4)**, and **Reviewer 1tu4 (R5)** provided valuable insights that help us improve the quality of our work.

Our work introduces MSFT, a multi-scale fine-tuning framework for encoder-based Time Series Foundation Models (TSFMs), which explicitly accounts for scale as a confounding factor and models both intra- and inter-scale dependencies. The method is simple to integrate and demonstrates consistent improvements across backbones and datasets.

We are especially encouraged that the reviewers acknowledged the strengths of our submission across all key dimensions.

* **Significance**: R1, R2, and R4 highlighted the timeliness and practical importance of fine-tuning TSFMs, addressing an urgent and underexplored challenge.

* **Quality**: R1, R2, R3, and R4 found the work well-executed, with rigorous and comprehensive experiments.

* **Empirical strength**:  R2, R3, R4, and R5 emphasized the robustness of our results.

* **Clarity**: R1, R4, and R5 appreciated the clear writing and logical structure.

* **Originality**: R1, R2, R4, and R5 recognized the novelty, soundness, and plausibility of our method.


More importantly, we sincerely appreciate the additional comments from the reviewers and have addressed them as follows:

* **Validation of overfitting** (R1) – Reported loss values at key checkpoints and will include full loss curves in manuscript.

* **Evaluation of knowledge forgetting** (R1) – Added transfer-learning results and clarified that modules can be disabled on unseen data to preserve pretrained knowledge.

* **Detailed pseudocode** (R2) - Provided to further improve the clarity.

* **Statistical robustness** (R2, R3) - Reported standard deviations from multiple runs, showing MSFT is robust to randomness.

* **More ablation study** (R3) - Added more thorough ablation experiments on in-scale masking.

* **Verification of claims** (R2, R3, R5) – Provided empirical validation to support our causal model and the identified challenges.

* **Related work** (R3, R4, R5) - Refined definition of multi-scale modeling and discuss fine-tuning for other TSFM architectures.

* **More efficiency analysis** (R5): Provided computational cost comparison with other fine-tuning approaches.


In the revised version, we will incorporate the reviewers’ comments to further improve our manuscript. Specifically, we will include loss curve figures, formal pseudocode, and a more detailed discussion of related work to better distinguish similar notions. The experiments will also be expanded to include the additional results introduced during the rebuttal, and the standard deviations will be reported in a dedicated section.

We hope this summary helps clarify how we have addressed the reviewers’ concerns and further improved the paper. We remain happy to engage in further discussion and sincerely appreciate your time and consideration.

Best regards,

The Authors

---

### Decision · Program_Chairs · 2025-09-17

**Decision:**

Accept (poster)

**Comment:**

The paper argues that current fine-tuning techniques for time series (TS) foundation models are suboptimal because they fail to exploit the inherent multi-scale nature of the data and the ability of foundation models to capture dynamics at multiple temporal resolutions. The authors propose a causal analysis of fine-tuning for TS forecasting and identify scale as a confounding factor that may induce spurious correlations in the trained model. To address this, they generate multi-scale inputs through downsampling. To effectively integrate information across scales, they introduce scale-specific adapters and decouple token dependencies by employing distinct attention mechanisms for within-scale and cross-scale information. Experiments are conducted on both deterministic and probabilistic forecasting tasks.


The reviewers agree on the relevance of the problem, the novelty of the multi-scale integration approach for fine-tuning pretrained foundation models, and the breadth of the experimental evaluation supporting the method. Their initial reviews, however, noted weaknesses, including an imprecise technical description, limited support for the causal analysis, and insufficient experimental evidence for certain claims. In their rebuttal, the authors provided extensive clarifications and additions, including new experiments (comparisons with baselines, ablations, and analyses), pseudo-code, and further explanations. The reviewers consider that most of their concerns have been addressed and hold a positive view of the paper. They encourage the authors to carefully incorporate the additional material into the main text and the appendix.